# Reprocessing of XBT profiles from the Ligurian and Tyrrhenian seas over the time period 1999-2019 with full metadata upgrade

Simona Simoncelli[1], Franco Reseghetti[2, §], Claudia Fratianni[1], Lijing Cheng[3,4], Giancarlo Raiteri[2]

[1] Istituto Nazionale di Geofisica e Vulcanologia (INGV), Viale Berti Pichat 6/2, 40127 Bologna, Italy, https://ror.org/029w2re51;

[2] Italian National Agency for New Technologies, Energy and Sustainable Economic Development (ENEA), S. Teresa Marine Research Centre, 19032 Pozzuolo di Lerici, Italy;

[3] International Center for Climate and Environment Sciences, Institute of Atmospheric Physics, Chinese Academy of Sciences, Beijing, 100029, China;

[4] Center for Ocean Mega-Science, Chinese Academy of Sciences, Qingdao, 266071, China;

[§] Now at Istituto Nazionale di Geofisica e Vulcanologia (INGV), Viale Berti Pichat 6/2, 40127 Bologna, Italy;

*Correspondence to*: Simona Simoncelli (simona.simoncelli@ingv.it)

## Abstract

The advent of open science and the United Nations Decade of Ocean Science for Sustainable Development are revolutionizing the ocean data sharing landscape for an efficient and transparent ocean information and knowledge generation. This blue revolution raised awareness on the importance of metadata and community standards to activate interoperability of the digital assets (data and services) and guarantee that data driven science preserve provenance, lineage and quality information for its replicability. Historical data are frequently not compliant with these criteria, lacking metadata information that was not retained crucial at the time of the data generation and further ingestion into marine data infrastructures. The present data review is an example attempt to fill this gap through a thorough data reprocessing starting from the original raw data and operational log sheets. The data gathered using XBT (eXpendable BathyThermograph) probes during several monitoring activities in the Tyrrhenian and Ligurian Seas between 1999 and 2019 have been first formatted and standardized according to the latest community best practices and all available metadata have been inserted, including calibration information never applied, uncertainty specification and bias correction from Cheng et al. (2014). Secondly, a new automatic Quality Control (QC) procedure has been developed and a new interpolation scheme applied. The reprocessed (REP) dataset has been compared to the data version, presently available from SeaDataNet (SDN) data access portal, processed according to the pioneering work of Manzella et al. (2003) conducted in the framework of the EU Mediterranean Forecasting System Pilot Project (Pinardi et al., 2003). The comparison between REP and SDN datasets has the objective to highlight the main differences derived from the new data processing. The maximum discrepancy among the REP and SDN data versions resides always within the surface layer (REP profiles are warmer than SDN ones) until 150 m depth, generally when the thermocline settles (from June to November). The overall bias and root mean square difference are equal to 0.002 ºC and 0.041 ºC, respectively. Such differences are mainly due to the new interpolation technique (Barker and McDougall, 2020) and the application of the calibration correction in the REP dataset.

The REP dataset (Reseghetti et al., 2024; https://doi.org/10.13127/rep_xbt_1999_2019.2) is available and accessible through the INGV (Istituto Nazionale di Geofisica e Vulcanologia, Bologna) ERDDAP (Environmental Research Division's Data Access Program) server, which allows machine to machine data access in compliance with the FAIR (Findable, Accessible, Interoperable and Reusable) principles (Wilkinson et al., 2016).

## 1 Introduction

The open science paradigm boosted the sharing of data through different pathways determining the generation of different versions of the same datasets. This might depend on the timeliness of data delivery, either in Near Real Time (NRT) or Delayed Mode (DM), the data center managing the dataset, the data assembly center or the marine data infrastructure collating it. The awareness of the importance of a complete metadata description is increasing among the scientific community since it allows interoperability, traceability of the data lifecycle, transparency and replicability of the knowledge generation process. In particular, some key information is crucial in climate science because it allows reanalysis of historical data, quantifying and reducing uncertainties, which are used to derive accurate scientific knowledge (Simoncelli et al., 2022).

The data provider should define the overall quality assurance strategy along with the data lifecycle to guarantee the availability of the best data product, which implies the possibility of reprocessing the dataset according to the state-of-the-art Quality Control (QC) procedures and standards. Data driven research should use the most extensive datasets with complete metadata information passed through a trustworthy QC procedure. These are also basic requirements to guarantee data reusability once the data are made openly accessible. The complete set of metadata assures transparency of the data provenance and avoids the circulation of multiple versions.

The integration in global databases of data not compliant with these principles emerged recently for measurements gathered in the last century, when the importance of storing data with complete ancillary information was not yet clear. A striking example is provided by the XBT (eXpendable BathyThermograph) probes, the oceanographic instruments that recorded the largest number of temperature profiles in the ocean from the 1970s to the 1990s (Meyssignac et al., 2019). The complete metadata information is crucial for QC, data reprocessing (Cheng et al., 2014; 2018; Goni et al., 2019) and integration with other data types to estimate key ocean monitoring indicators, such as the trend of global ocean heat content (Cheng et al., 2020; 2021; 2022), one of the most important climate change indicators. According to the literature (Cheng et al., 2016 and 2017; Parks et al., 2022), the crucial metadata information that must be associated with XBT data includes probe type and manufacturer, fall rate equation, launch height, and recording system. This information was not mandatory for the data ingestion in the main marine data infrastructure, thus most historical data miss it. For example, 50% of XBT profiles in the World Ocean Database (WOD) have no information about manufacturer or probe type (Cowley et al. 2021), necessitating the application of intelligent metadata techniques to complement it (Palmer et al., 2018; Leahy et al., 2018; Haddad et al., 2022).

This data review originated from the recognition that the historical XBTs from the Ligurian and Tyrrhenian Seas, presently available in the main marine data infrastructures - SDN (https://www.seadatanet.org/), WOD

([https://www.ncei.noaa.gov/products/world-ocean-database](https://www.ncei.noaa.gov/products/world-ocean-database)),      Copernicus      Marine      Service      (CMS,

[https://marine.copernicus.eu/](https://marine.copernicus.eu/)) - have incomplete metadata description and the data might also differ. Our

objective was to recover the raw data together with the full metadata description and secure them to the future

generation of scientists for their further use. This awareness raised contemporary to the evolution of open

science and FAIR (Findable, Accessible, Interoperable and Reusable) data management principles, which

motivated us to adopt the latest community standards, QC procedures, and to implement an ERDDAP server

as data dissemination strategy. ERDDAP is an open source environmental data server software developed by

NOAA and used throughout the ocean observing community (Pinardi et al. 2019; Tanhua et al. 2019) which

allows us to become a node of the present data digital ecosystem, in line with one of the expected societal

outcomes ("transparent and accessible" ocean) of the UN Decade of Ocean Science 2021-2030 (Ryabinin et

al., 2019; Simoncelli et al., 2022).

The paper describes the reprocessing of temperature profiles from expendable probes deployed between 1999

and 2019 in the Ligurian and Tyrrhenian seas, most of them from vessels operating  a commercial line between

the Italian ports of Genova and Palermo within the Ships Of Opportunity Program (SOOP) of the Global Ocean

Observing System (GOOS), currently identified as MX04 line. Additional XBT data were collected through

ancillary monitoring surveys with commercial and research vessels. The dataset contains some XCTD

(eXpendable Conductivity-Temperature-Depth probes) profiles (less than 1%) too. The reprocessed dataset

(REP) is obtained from the original raw XBT profiles, the readable output of the Data Acquisition System

(DAQ). A correction based on the DAQ calibration (when available) is applied to each temperature recorded

value but also provided as separate information, to allow the user to eventually subtract it. Automated QC

tests, specifically tuned for western Mediterranean basins, based on the latest documented QC procedures

(Cowley et al., 2022; Parks et al., 2022; Good et al., 2023; Tan et al., 2023) and best practices to assign a

Quality Flag (QF) are applied, followed by interpolation of raw profiles at each meter depth. All available

information collected during data-taking has been added in the metadata section, according to the SeaDataNet

standards ([https://www.seadatanet.org/Standards](https://www.seadatanet.org/Standards)) and IQuOD (International Quality-controlled Ocean

Database, [https://www.iquod.org/index.html](https://www.iquod.org/index.html)) recommendations. Uncertainty specification for both depth and

temperature is also provided, being a crucial information for assimilating data in ocean reanalysis or for

utilizing them in downstream applications. Cheng et al. (2014) demonstrated that XBT data are characterized

by systematic bias when compared with data gathered from CTD, and computed the commonly used correction

scheme for both temperature and depth records, which is very important to derive integrated data products or

ocean indicators from multiple data sources and instruments (Cheng et al., 2016). The REP dataset includes

Cheng et al. (2014) correction scheme applied to the calibrated profiles at original depth and then interpolated

at each meter depth.

The REP data product allows the user to select from the original profiles to the validated, interpolated and

corrected ones, filtering on the basis of the required quality level, selecting the associated QF. Furthermore,

the dataset is accessible through the ERDDAP (Environmental Research Division's Data Access Program) data

server (http://oceano.bo.ingv.it/erddap/index.html) installed at INGV (https://ror.org/029w2re51) which provides a simple and consistent way to download it in several common file formats.

This study was conducted in the framework of the MACMAP (Multidisciplinary Analysis of Climate change indicators in the Mediterranean And Polar regions) project (https://progetti.ingv.it/it/progetti-dipartimentali/ambiente/macmap) funded by INGV (https://ror.org/00qps9a02) (2020-2024) in technical collaboration with ENEA (Italian National Agency for New Technologies, Energy and Sustainable Economic Development) and GNV (Grandi Navi Veloci) shipping company. In fact, the reprocessing of the historical XBTs was preparatory to the automatic validation, management and publication of new XBT data gathered on the MX04 line from September 2021, after two years interruption of the monitoring activity.

The paper is organized as follows: Section 2 describes the main characteristics of an XBT system; Section 3 describes the original dataset and the monitoring activities that sustained it; Section 4 describes the methodology applied for the automatic QC and the correction derived from calibration; Section 5 is about the results; Section 6 summarizes the main results and draws conclusions; Section 7 describes the REP dataset findability and accessibility.

## 2 The XBT system

In the early 1960s, following a request from the US Navy looking for a seawater temperature profiler for military applications, engineers from Francis Associates developed an early version of an XBT probe. The prototype was improved within Sippican Corp. (now part of Lockheed Martin Co., hereinafter Sippican) and then adopted by the US Navy (Reid, 1964; Arthur D. Little, 1965 and 1966). Within a few years Sippican optimized the original project and marketed different XBT types with specifications suitable for various depths and ship speed. XBTs became very popular within the oceanographic community (Flierl and Robinson, 1977) allowing the gathering of Temperature (T) profiles through the use of commercial vessels (ships of opportunity) and not just research vessels.

The XBT system consists of: an expendable ballistic probe falling into seawater; a device (DAQ) that records an electrical signal and converts it into usable numerical data (in combination with a computer unit) and the connection between the falling probe and the DAQ (e.g. Goni et al., 2019 and Parks et al., 2022). The sensing component is an NTC (Negative Temperature Coefficient) thermistor that changes its resistance according to the temperature of seawater flowing through the central hole of the probe nose where it is located. Its thermal time constant $\tau$ (time needed to detect 63% of a thermal step signal) is $\sim 0.11$ s (Magruder, 1970 and references therein) so a time of ~0.6 s is needed to detect a step temperature change. Technical characteristics required by Sippican for the NTC thermistor, reading circuit and resistance to temperature conversion procedure (e.g. Sippican 1991 and Appendix A), put some limits on the accuracy of XBT measurements.

Another essential component is the thin twin copper wire which is part of the acquisition circuit and which is unwound by two spools simultaneously (clockwise from the ship and counterclockwise from the falling probe), a technique which decouples the XBT vertical motion from the translational motion of the ship. The albeit weak electric current that runs through the wire during acquisition transforms the wire into a large antenna

sensitive to nearby electromagnetic phenomena. A non-uniform coating application and a defective winding
on one of the spools cause a significant part of the faulty or prematurely terminated acquisitions.

XBT probes do not house any pressure sensor and the depth associated with a temperature measurement is not
measured directly but estimated by a Fall Rate Equation (FRE) provided by the manufacturer with coefficients
that depend on the probe type and are valid for the world ocean. The software transforms a time series of
resistance values sensed by the thermistor into a series of depth - T values using first a resistance-to-
temperature conversion relationship (identical for all XBT types because it is specific for the thermistor used,
see Appendix A) and then calculating the corresponding depth values by applying a specific FRE for each
probe type. Sippican has preset conservative values for the recording time in its acquisition software but these
values can be freely modified in order to use all the wire wound on the probe spools. The first column of Table
1 shows the nominal values and the maximum recorded depth in the same areas for each specific probe type.

Each component of an XBT system contributes to the overall uncertainty on depth and T measurements.
Recently the IQuOD group (Cowley et al., 2021) released a summary of T uncertainties specifications for
different oceanographic devices determined using available knowledge (Type B uncertainty). The uncertainty
estimate associated with XBT probes adopts the accuracy values provided by the manufacturer:

• for depth: 4.6 m up to 230 m depth and 2% at greater depths;

• for T: within the range 0.1 - 0.2 °C, with small variations depending on the manufacturer and the
manufacturing date. The value associated with the XBT probes in the REP dataset is equal to 0.10 °C.

Bordone et al. (2020) compared XBT profiles from SOOP activities in the Ligurian and Tyrrhenian Sea with
quasi contemporaneous (± 1 day) and co-located (distance smaller than 12 km) Argo profiles. The XBT
profiles used by Bordone et al. (2020) are included in the REP dataset but they went through a different QC
and interpolation procedure that could slightly modify their results. In the 0-100 m layer, the mean T difference
was 0.24 °C (the median 0.09 °C) and the Standard Deviation (SD) was 0.67 °C. Below 100 m depth, the XBT
measurements were on average 0.05 °C warmer than the corresponding Argo values (mean and median were
almost coincident) and the SD was 0.10°C. This last SD value agrees with the manufacturer specification and
the T uncertainty value reported by Cowley et al. (2021), which has been assigned to the REP data. The values
estimated by Bordone et al. (2020) for the surface and sub-surface layer (depth < 100 m) are instead affected
by both the XBT (4.6 m) and Argo (2.4 dbar) depth uncertainty estimation, meaning that a small variation in
depth could correspond to a large variation in temperature especially when the seasonal thermocline develops,
so that the comparison with Argo values would not be significant. The specified uncertainties are independent
of the systematic error or bias affecting the XBT temperature and depth measurements, that have been
corrected in the REP dataset applying the Cheng et al. (2014) correction scheme.

In fact, the first part of the XBT motion is critical, meaning that the T and depth values in the surface layer
must be considered very carefully, especially if the launch height (which influences the entry velocity of the
probe and consequently the time and depth at which it reaches the terminal velocity, i.e. the value used in the

FRE) differs from 3 m above sea level, the value suggested by Sippican. Very high launch platforms make the initial depth values calculated through the FRE incorrect (Bringas and Goni, 2015 and references therein). In addition, the time constant of the thermistor (Magruder, 1970 and references therein), the thermal mass of the XBT probe (e.g. Roemmich and Cornuelle, 1987) and the storage temperature, influence the reliability of the first T records. For these reasons, careful data validation in the near surface layer and where the seasonal thermocline occurs (i.e. depths shallower than 100 m in the study region), is crucial.

The depth resolution depends both on DAQ sampling rate and FRE of the XBT probe. All DAQ models used in this dataset work at 10 Hz (i.e. a sample every 0.1 s, a time interval nearly coincident with the time constant of the NTC thermistor) so that the depth resolution has actual values close to 0.6 m. The T resolution is usually 0.01 °C when using the standard Sippican software while 0.001 °C is the standard output for Devil/Quoll DAQs and some old Sippican software versions. Throughout the work, three decimal digits are always used for T values and the derived quantities (i.e. vertical gradient). The computer clock (always updated to the UTC value shortly before the start/after the end of operations) provides the time coordinate of each profile with a sensitivity of 1 s. The differences recorded with respect to the standard UTC time have always been smaller than 1 s over a 24 hour time frame.

Sippican's manuals released over the years (e.g. Sippican 1968, 1980, 1991, 2006, 2010 and 2014) and reports (e.g. Sy, 1991; Cook and Sy, 2001; Sy and Wright, 2001; Parks et al., 2022) well describe the best practices for XBT use. The checking of the XBT system with a tester before and after data collection as well as the complete description of the system characteristics in the metadata is highly recommended for an optimal use of XBT measurements. When strip chart recorders were used, a preliminary and accurate calibration of the acquisition unit with a tester was mandatory (e.g. Sippican, 1968 and 1980; Plessey-Sippican, 1975). With the advent of digital systems this procedure was also recommended (Bailey et al., 1994). Only since July 2010 the tester check has been introduced in the monitoring activity along the MX04 line and few other subsets of profiles contained in the REP dataset. Reseghetti et al. (2018) found a reduction of the (XBT-CTD) temperature difference after introducing a correction based on the tester check. This was also confirmed by the comparison between XBT and Argo profiles described in Bordone et al. (2020). Based on these findings, a specific correction has been developed and it represents a key component of the information never used in previous data versions and unlocked in the REP dataset (section 4.3).

The first XCTD models were developed by Sippican (Sippican, 1983) in the 1980s and were analog. They were completely replaced in the last years of the last century by digital versions produced by the Japanese company TSK (Tsurumi Seiki Co.). XCTD-1 probes present some differences compared to XBTs in terms of resolution and accuracy, and a completely different recording circuitry. The manufacturer (the Japanese company TSK) claims an accuracy of 0.02 °C on T (a factor of five better than XBTs) and a resolution of 0.01 °C while the depth accuracy is the same as for XBT probes. These accuracy values can be considered Type B uncertainties, as in Cowley at al. (2021), and they are included in the REP dataset metadata information. The sampling frequency is 25 Hz (i.e. a reading of the thermistor resistance value every 0.04 s) with a falling speed

which is just over half that the XBT probes (see Table 1), the depth resolution for the model XCTD-1 is about 0.14 m.

## 3 The dataset

3782 temperature profiles, collected from September 1999 to September 2019 in operations managed by ENEA (S. Teresa Marine Research Centre, STE thereafter) mainly through the use of commercial ships, are included in the REP dataset. They come from XBT probes, plus a few dozen XCTDs. Figure 1 shows the XBT profiles temporal and spatial distribution, highlighting their sparseness, mainly influenced by the irregular monitoring activity and data concentration along the MX04 Genova-Palermo line. The vertical data distribution (Figure 1c) is also non-homogeneous due to the local bathymetry, the use of different probe types and the ship speed.

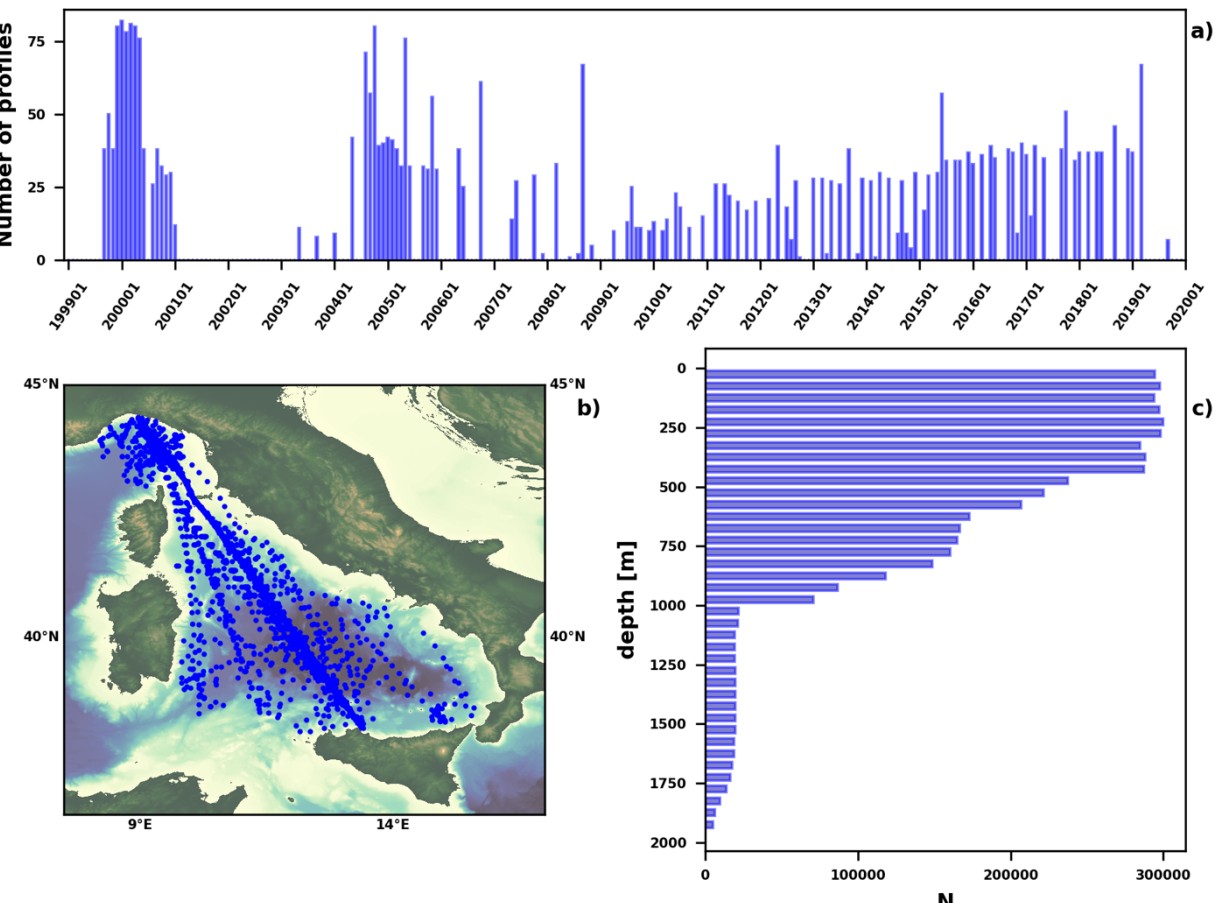

**Figure 1 (a) temporal distribution of the REP (reprocessed) XBT profiles; (b) geographical location; (c) vertical distribution in layers of 50 m of depth.**

Table 1 shows some of the characteristics of the expendable probes used in this dataset, the FRE coefficients applied to calculate the depth and the mass of the various components of each probe type (ZAMAK - Zink Aluminium Magnesium Kupfer - for the nose, plastic for the body and spool and copper wire, considering the total quantity that can unwind from the on-board spool), which allows to evaluate the overall quantity of

material abandoned at sea caused by the REP dataset. We have no information regarding the components of the XCTD-1 probes but their nose is made of plastic material. Sippican is the manufacturer of all the XBT probes used, while the XCTD-1 probes are manufactured by TSK - Tsurumi Seiki Co. and marketed in Italy by Sippican.

The profiles were gathered during the following monitoring activities:

1. SOOP monitoring on the Genova-Palermo MX04 line, which provides the greatest contribution both in terms of campaigns (1999-2000, 2004-2006, 2010-2019) and quantity of profiles;
2. SOOP monitoring in collaboration with CSIRO (Commonwealth Scientific and Industrial Research Organization), from 2007 to 2011;
3. Sporadic additional SOOP monitoring by ENEA-STE in the Mediterranean (2012-2014);
4. An agreement between ENEA and IIM (Italian Hydrographic Institute of the Navy), (2006 - 2019);
5. An operational collaboration between ENEA-STE and National Research Council of Italy - Institute of Marine Sciences (CNR-ISMAR, Lerici), (2000 - 2017).

The main characteristics of the vessels and the instrumentation used for the data collection are summarized in Appendix B.

**Table 1 Characteristics of the different probes used: nominal depth suggested (and guaranteed) by Sippican and experienced maximum depth in the Mediterranean; maximum ship speed suggested by Sippican for an optimal drop; coefficients of Fall Rate Equation $D(t) = At - Bt^2$ used for depth calculation (provided by the manufacturer or by IGOSS, Hanawa et al., 1995); per probe amount of ZAMAK, copper and plastic and the number of probes included in the dataset for each probe type.**

| Probe type | Rated depth (max depth) (m) | Rated ship speed (knots) | Coeff. $A$ ($ms^{-1}$) | Coeff. $B$ ($ms^{-2}$) | ZAMAK (kg) ± 0.001 | Plastic (kg) ± 0.001 | Copper (kg) ± 0.002 | REP dataset |
|---|---|---|---|---|---|---|---|---|
| T4 | 460 (583) | 30 | 6.691 | 0.00225 | 0.613 | 0.052 | 0.202 | 1436 |
| T5 | 1830 (2272) | 6 | 6.828 | 0.00182 | 0.613 | 0.125 | 0.357 | 61 |
| T5/20 | 1830 (2248) | 20 | 6.828 | 0.00182 | 0.613 | 0.125 | 0.726 | 188 |
| T6 | 460 (588) | 15 | 6.691 | 0.00225 | 0.613 | 0.052 | 0.158 | 69 |
| T7 | 760 (977) | 15 | 6.691 | 0.00225 | 0.576 | 0.052 | 0.240 | 61 |
| DB | 760 (962) | 20 | 6.691 | 0.00225 | 0.576 | 0.052 | 0.294 | 1759 |
| T10 | 200 (292) | 10 | 6.301 | 0.00216 | 0.613 | 0.052 | 0.098 | 173 |
| XCTD-1 | 1100 (1100) | 12 | 3.425432 | 0.00047 | None | NA | 0.440 | 35 |

The first SOOP in the Mediterranean Sea (September 1999 - December 2000) started in the framework of the European Mediterranean Forecasting System Pilot Project (MFSPP, Pinardi et al., 2003; Manzella et at., 2003; Pinardi and Coppini, 2010) under INGV coordination to support the development of operational oceanography

forecasting activities through the NRT provision of ocean observations. XBT profiles were collected along transects crossing the Mediterranean Sea designed to monitor the variability of the main circulation features. The raw profiles were subsampled on board by Argos software (15 inflection points) and quickly inserted into the Global Telecommunication System (GTS) while the full resolution profiles were sent to the ENEA-STE assembly center for QC, interpolation and NRT provision to the forecasting center (e.g. Fusco et al., 2003; Manzella et at., 2003; Zodiatis et al., 2005; Millot and Taupier-Letage, 2005a and 2005b). The MX04 line is the only SOOP line still active in the Mediterranean Sea on seasonal basis, thanks to the MACMAP project and the collaboration with GNV, whose ships connect daily (just under 20 hours sailing at about 22 knots) Genova (44.40 °N, 8.91 °E) to Palermo (38.13 °N, 13.36 °E).

Starting from September 1999, 20 campaigns were carried out, in collaboration between CNR-ISMAR and ENEA-STE, with initial monthly monitoring frequency, then every 15 days (December 1999 - May 2000), and again monthly frequency until December 2000. T4 probes (with some T6 probes) were launched at fixed intervals of time (every 30 minutes), corresponding to a sampling distance of about 11 nm. A Sippican MK12 card inserted into the motherboard of a desktop running Windows 98 IIE and with the software set to stop acquisition at 460 m depth was used. All the campaigns were carried out using the MV "Excelsior", its route was always the same and almost coincident with track 44 of the altimetric satellites (Vignudelli et al., 2003). After a hiatus of more than 3 years and a campaign in May 2004 to check slightly different operational procedures, monitoring along the MX04 line resumed on a monthly basis from September 2004 to December 2005 (no cruises in July and August 2005), with two additional cruises in May and October 2006, for a total of 17 campaigns within the EU MFS-Toward Environmental Prediction project (MFS-TEP, Manzella et al. 2007; Pinardi and Coppini, 2010). The ships (always GNV vessels) followed a route with marginal differences compared to the previous one due to the introduction of nature conservation limitations in the Tuscan archipelago. In November 2004, February and December 2005 the route was significantly different due to bad weather and sea conditions. The campaigns were planned to travel as close as possible to the passage date of the Jason-1 altimetric satellite along track 44 and for this reason some were carried out on the route traveled in the opposite direction, independently on weather and sea conditions. T4 and DB XBT probes were usually deployed (with a few XCTD-1 and some T6) and the sampling distance was variable from 8 to 12 nm. After a few months, the DAQ (a Sippican MK21 ISA), despite excellent operating conditions and good ground connection, began to record profiles with rapid oscillations (amplitude $\simeq 0.05$ °C) not attributable to the known water masses characteristics (not shown). Only at the end of the MFS-TEP data taking, careful laboratory checks identified a pair of capacitors on the ISA board as responsible for this malfunction. Unlike MFS-PP, the acquisition software was set to use all the wire available on the probe spool (i.e. 600 m for T4 and 1000 m for DB probes).

Monitoring on the MX04 line resumed in July 2010, managed directly by ENEA-STE and until January 2013 was widely variable both in terms of frequency and sampling distance (due to the uncertainty in the supply of XBT probes). A regular sampling scheme was then adopted with a launch every 10' of latitude (corresponding to 11-12 nm depending on the ship's course), excluding the archipelago of Toscana, with five to six annual

repetitions, following the same route as in 2004-2006 (excluding February 2013 and April 2014 because of very bad weather and sea conditions). It was also decided to carry out monitoring campaigns only with good weather and sea conditions. From June 2015, the ships moved to a more westerly route in the northern part of the transect crossing the Corsica Channel (this allows monitoring of the water exchange between the Tyrrhenian Sea and the Ligurian Sea) to rejoin the previous one around at latitude 39°N. The number of drops at fixed positions increased to thirty-seven, mainly DB probes while other XBT types were used in particular areas due to the reduced bathymetry (T10) or with interesting deep thermal structures (T5/20). Based on the experience from XBT vs. CTD comparison tests, since March 2011 the XBT probes were placed in the open air (but always in the shade) for at least half an hour before the deployment to allow them to thermalize with the atmosphere and reduce as much as possible the temperature difference with the sea surface layer.

A short SOOP activity in collaboration with CSIRO was completed between December 2007 and March 2011 (19 campaigns) using containerships from Hapag Lloyd (namely "Canberra Express", "Stadt Weimar" and "Wellington Express") and CMA CGM ("CMA CGM Charcot") shipping companies, operating between Northern European ports and Australia. These campaigns were characterized by irregular frequency throughout the year, a very high launching platform (25 m over the sea level or more) and a sampling distance between 20 and 35 nm. XBT launches began near the Egadi Islands (west of Sicilia) and terminated in the Corsica Channel, following a path halfway between the MX04 transect and the island of Sardinia. CSIRO installed a Turo Devil DAQ on each vessel while ENEA-STE provided the DB probes.

Some additional XBT profiles (mainly DB type) were gathered in the Ligurian Sea between May 2012 and March 2014 on board the GNV ship "Excellent" (in 5 campaigns) and in 2014 two different cruises using a Sippican MK21 USB onboard the container ship "Daniel A" from the Turkish shipping company ARKAS.

From 2006 to 2019, 10 campaigns were carried out in collaboration between ENEA and IIM, using the ships "Ammiraglio Magnaghi", "Aretusa" and "Galatea", collecting a total of about 200 profiles using different XBT types, deployed from different heights and using different DAQs.

Finally, an operational collaboration between ENEA-STE and CNR-ISMAR allowed to carry out 29 campaigns between 2000 and 2017 using vessels managed by the CNR (mainly RV "Urania", but also RV "Minerva Uno" and "Ibis"), gathering several hundred profiles with different XBT probe types deployed from different heights and recorded using four different Sippican DAQ units.

The total amount of material abandoned at sea, due to the launch of the XBT/XCTD probes which constitute the REP dataset, is provided using the per-probe values reported in Table 1: over 2300 kg of ZAMAK, 220 kg of plastic material and 1060 kg of copper wire. Furthermore, there was no additional contribution to greenhouse gas emission since mainly commercial vessels were used and, in the case of research vessels, the launch of XBT probes was ancillary to the main activities of the cruise.

## 4 Methodology

Specific QC procedures for XBT profiles in the Mediterranean Sea were first developed by Manzella et al. (2003) within the MFS-PP project and later improved in Manzella et al. (2007). Temperature observations in

the Mediterranean Sea, due to its thermohaline circulation, water mass characteristics and large temperature variability, might present peculiar features like thermal inversions or zero thermal gradient in areas of deep water formation, thus necessitating regional tuning of QC tests. The prior QC procedures included: detection of profile's end, gross range check, position control, elimination of spikes, interpolation at 1 m intervals, Gaussian smoothing, general malfunctioning control, comparison with climatology and final visual check by operator. Some additional constraints were applied: elimination of the initial part of each profile (the first acceptable value is at 4 m depth, following the standard international procedure), allowed temperature values within the 10-30 °C interval, maximum temperature inversion of 4.5 °C in the 0-200 m layer, 1.5 °C below 200 m, and 3 °Cm$^{-1}$ as maximum thermal gradient. This QC has not been applied to the data released in NRT through the GTS (Global Telecommunication System, https://community.wmo.int/en/activity-areas/global-telecommunication-system-gts) but only to the data made available in DM through the SDN infrastructure (accessible through the relative saved query from the SDN CDI data access portal at https://cdi.seadatanet.org/search/welcome.php?query=1866&query_code={4E510DE6-CB22-47D5-B221-7275100CAB7F}). The raw data for the GTS dissemination were provided to NOAA and in the early 2000s the profiles were also heavily sub-sampled due to the low bit rate satellite system provided by Argos, the basic GTS data transmission system (Manzella et al., 2003). These different dissemination channels contributed to the existence of several versions of the same profile in different blue data infrastructures (i.e. WOD, SDN).

A new automated QC procedure, written in Python and structured as a package, has been implemented in the framework of the MACMAP project starting from the original raw XBT profiles, considering the scientific progress made in the field in the last two decades and the full metadata information available. The aim was twofold: first to secure the best version and most complete dataset for further use to the scientific community; secondly to implement an automated QC workflow for the seasonal XBT campaigns started in September 2021 thanks to the MACMAP project. This also allowed to refine and standardize the quality assurance procedures on board of the vessels to record all ancillary information in a pre-defined format and minimize the impact of different operators on the data quality. The calibration correction, detailed in section 4.3, has been added, when available, to the raw data before the QC analysis. However, it is provided as a separate variable associated with each XBT profile and the user can remove it, if required. None of the original data has been deleted but integrated with quality indexes, with the exception of those repeated during data taking. These replicates have been decided by the operator during the sampling activity when the observed profile was affected by serious acquisition problems, both external (i.e. electrical discharge) and probe-specific (wire break or anomalous stretching, insulation penetration, leakage and so on).

A final visual check has also been performed using ODV software (R. Schlitzer, Ocean Data View, https://odv.awi.de/, 2023) which highlighted the presence of anomalous behavior in some T profiles that the automatic QC tests could not detect. Some examples will be discussed in Section 5 (Figure 10). This visual check suggested assigning to each profile a general QF, choosing between these two options: 1) *excellent* indicating all QC done and 2) *mixed* indicating some problems, with comments to warn the user about the anomalous features.

## 4.1 Automatic Quality Control procedure

The XBT raw profiles have been QCed using a sequence of independent tests, checking for invalid information on geographic characteristics and for known signatures of spurious measurements. Results of each test are recorded by inserting the relative exit values to the corresponding measurement in ancillary variables (POSITION_SEADATANET_QC, DEPTH_TEST_QC, TEMPET01_TEST_QC) according to the scheme shown in Table 2, while Figure 2 provides an example of the QC tests applied to a profile.

The independent QC tests are described hereafter.

**Position on land check**

The profile position should be located at sea, thus latitude and longitude of each profile is checked against gridded GEBCO bathymetry (GEBCO Compilation Group, 2022) on a 15 arc-second interval grid to determine if it is located on land or not (test 1): if the "height" is negative it is lower than sea level, and it is flagged as GOOD ('profile is at sea'), otherwise is flagged as BAD ('profile is on land'). The ancillary variable, POSITION_SEADATANET_QC, contains the exit value of the position check. However, there are no data flagged as BAD due to position on land in the REP dataset, since the operators checked both the position and the launch time before the data transmission to the data assembly center (ENEA-STE). Since we did not encounter specific issues with date and time we did not implement additional checks.

**Depth check**

The depth values of each XBT profile are compared to the *local bottom depth* extracted from GEBCO (test 2) and the *last good depth* (test 3) value provided by the operator. Depth values are flagged as GOOD ('depth is below reference depth value') if they are shallower than it otherwise they are flagged as BAD ('depth is above reference depth values'). The corresponding local bottom depth extracted from GEBCO (BATHYMETRIC_INFORMATION) and the last good depth value provided by the operator (LAST_GOOD_DEPTH_ACCORDING_TO_OPERATOR) are annotated in the metadata as global attributes associated to each profile to facilitate further analysis by expert users.

**Table 2 Summary of the automated QC tests, the assigned exit values to each measurement and the ancillary variables containing them.**

| Test | Check | Description | Exit value | Exit value description | Ancillary variable |
|------|-------|-------------|-----------|------------------------|--------------------|
| 1 | Position control | Function to detect incorrect longitude and latitude values | 1/4 | 1 coordinates at sea<br>4 coordinates on land | POSITION_SEADATANET_QC |
| 2 | Depth | Function to detect depth values out of extreme depths. The reference depth is the **local bottom depth** from GEBCO. | 1/4 | 1 depth is below reference depth<br>4 depth is above reference depth | DEPTH_TEST_QC |
| 3 | Depth | Function to detect depth values out of extreme depths. The reference depth is the **depth indicated by the operator**. | 1/4 | 1 depth is below reference depth<br>4 depth is above reference depth | DEPTH_TEST_QC |
| 4 | Gross range check | Function to detect T values out of ranges in Table 3 | 49/52 | 49: T inside the range<br>52: T is out of range | TEMPET01_TEST_QC |
| 5 | Surface | Function to flag the first 4 | 49-52 | 49: T difference < 1 SD | TEMPET01_TEST_QC |

| # | Name | Description | Range | Exit values | QC variable |
|---|------|-------------|-------|-------------|-------------|
| | | meters considering as reference std=0.1 and its growing | | 50: 1 SD < T difference < 2 SD<br>51: 2 SD < T difference < 3 SD<br>52: T difference > 3 SD | |
| 6 | Vertical gradient | Function to detect stuck values, decreasing and increasing values according to gradient value and considering only the values that passed the previous checks | 56-58 | 56: stuck value<br>57: negative gradient out of threshold<br>57#: negative gradient out of threshold in successive iteration (#=1 or 2)<br>58: positive gradient out of threshold<br>58#: positive gradient out of threshold in successive iteration (#=1 or 2) | TEMPET01_TEST_QC |
| 7 | Wire break/ stretch | Function based on vertical gradient check to identify wire break on shipside or on probe-side | 61 | 61: wire break/stretch | TEMPET01_TEST_QC |
| 8 | Spike detection | Function to detect spike considering the median, media and thresholds $s_k$ in Table 4 | 59 | 59: spike if $|T3-median(T1,T2,T3,T4,T5)| \neq 0$ and $|T3-mean(T1,T2,T3,T4,T5)| > s_k$ | TEMPET01_TEST_QC |
| 9 | High Frequency spiking | Function to identify feature in the profile like critical drops | 60 | 60: critical drop | TEMPET01_TEST_QC |

**Gross range check**

The Gross range check applies a gross filter on observed temperature considering T thresholds that vary on 5

vertical layers, as reported in Table 3. T thresholds have been defined analyzing the seasonal T distribution in

4 sub-regions displayed in Figure 3: 1) the Ligurian Sea; 2) the Northern Tyrrhenian Sea; 3) the South-West

Tyrrhenian Sea; 4) the South-East Tyrrhenian Sea. The domain subdivision is based on the mean circulation

features at 15 m and 350 m depth, computed from the Mediterranean Sea reanalysis (Simoncelli et al., 2014)

data over the time period 1999-2018 (Figure 3). A detailed description of the circulation is out of scope here

but its main features are detailed in Pinardi et al. (2015) and von Schuckmann et al. (2016, section 3.1).

**Surface check**

In general, a probe needs a couple of seconds from the impact with the sea surface to stabilize its motion and

reach the terminal velocity (Bringas and Goni, 2015 and references therein). Different approaches have been

followed over the years on how to handle the near-surface values. In the late 70s, IOC proposed to extrapolate

upward isothermally the values from 3 to 5 m to obtain the surface temperature for encoding (IOC, 1975) while

the FNWC (U.S. Fleet Numerical Weather Central) procedure was to extrapolate from 8 feet (2.4 m) to the

surface using the slope at that depth. Wannamaker (1980) suggested reaching the surface starting from 4 m

using the slope between 4 and 6 m depth. Afterwards, other authors decided to discard the initial measurements,

considering only the values starting from a certain depth to be valid, also depending on the used DAQ (e.g.

Bailey et al. 1994; IOC, 1997; Kizu and Hanawa, 2002; Gronell and Wijffels, 2007; Cowley and Krummel,

2022 and reference therein). For example, Manzella et al. (2003) selected the value at 5 m depth as the first

acceptable value during MFS-PP project then changed to 4 m during MFS-TEP.

It is preferred that the user is provided all the original measurements by adding a test that analyzes the

measurements in the surface layer and annotating the resulting exit value in the ancillary variable. The

proposed test chooses as reference the value recorded at time t = 0.6 s (the first value currently considered

acceptable), calculates the differences between this value and shallower measurements and classifies them using the T standard uncertainty (SD) associated to an XBT probe (0.10 °C) as a metric. In detail, the temperature differences $T(t_{0.6})-T(t_i)$, with $(0.0 \leq t_i \leq 0.5)$ s are calculated and the QF is assigned as follows:

- GOOD if $|T(t_{0.6})-T(t_i)| \leq 1*SD$;
- PROBABLY GOOD if $1*SD<|T(t_{0.6})-T(t_i)| \leq 2*SD$;
- PROBABLY BAD if $2*SD<|T(t_{0.6})-T(t_i)| \leq 3*SD$;
- BAD if $|T(t_{0.6})-T(t_i)| > 3*SD$.

The flag GOOD means a value indistinguishable from the record at $t = 0.6$ s while PROBABLY GOOD defines an excellent compatibility. The PROBABLY BAD and BAD flags simply indicate a difference greater than the established threshold with respect to the reference value at $t = 0.6$ s.

**Inversion and gradient checks**

This test is performed to detect unrealistic T oscillations with abrupt T reversals or unusually large T gradients. The vertical gradient is defined as the difference between vertically adjacent measurements, $Tz=(T_2-T_1)/(Z_2-Z_1)$, where $T_2$ and $T_1$ are temperatures at depths $Z_2$ and $Z_1$, with level 2 being deeper than level 1. This test is applied three times iteratively discarding values that failed the test in the next iteration. The acceptable T gradient ranges (Table 3) have been defined through a statistical analysis in 5 vertical layers and 4 sub-regions (Figure 3) through an approach that blends expert decisions with statistical support. Due to the spatial (horizontal and vertical) and temporal sparseness of the data, the 0.01% and 99.99% quantiles have been computed in the 5 layers considering: 1) the whole dataset; 2) the 4 sub regions; 3) the entire domain but for 4 seasons. The thresholds are the absolute minimum 0.01% quantile and maximum 99.99% quantile deriving from the three cases. The thresholds of the two deepest levels are from case 1, the upper layer uses values from case 2 and the second and third layers use the results of case 3.

**Table 3 Temperature and thermal gradient thresholds defined in 5 layers.**

| Layer | Temperature (°C) | | Vertical Gradient (°Cm$^{-1}$) | |
|---|---|---|---|---|
| **0-100 m** | 12.000 | 30.000 | -3.400 | 0.613 |
| **100-250 m** | 12.500 | 17.900 | -0.317 | 0.244 |
| **250-450 m** | 12.700 | 15.500 | -0.156 | 0.170 |
| **450-1000 m** | 13.100 | 14.800 | -0.133 | 0.137 |
| **1000-2300 m** | 13.100 | 14.000 | -0.094 | 0.090 |

**Wire break/stretch**

Results of inversion and gradient checks are used to identify sharp variations toward negative values, indicating that the copper wire breaks on shipside, or toward high values (close to 35 °C or more), when the wire breaks

on probe-side where there is often a progressive increase in temperature values rather than a step transition to

full scale.

**Spike detection**

This test looks for single value spikes and it checks T measurements for large differences between adjacent

values. A spike is detected by computing the median value ($Med_k$) in a 5 points interval (3 m approximately)

with the profile value at the central point of the interval ($T_k$). The spike is detected and the consequent flag is

applied if $T_k$ is not equal to $Med_k$ and the difference ($s_k$) between $T_k$ and the mean ($Ave_k$) in the chosen

interval is greater than a threshold value.

$$Med_k = median(T_{k-2}:T_{k+2})$$

$$Ave_k = mean(T_{k-2}:T_{k+2})$$

$$s_k = T_k - Ave_k \quad , \quad c_k = T_k - Med_k \neq 0$$

The spike threshold values have been defined for the entire region in 5 vertical layers as the 99.9% quantile of

the $s_k$ distribution and they are reported in Table 4. Figure 4a shows the probability distribution of $s_k$ values

with $c_k$ not equal to zero in 5 layers. $s_k$ distribution is characterized by large values above 80 m that diminish

with depth, as the temperature variability does. The $s_k$ scatter plot (Figure 3b) shows its values along the water

column, with the red dots highlighting the values over the selected thresholds.

**Table 4 Spike detection threshold defined in 5 vertical layers.**

| Layer | spike threshold (°C) |
|:---:|:---:|
| 0-80 m | 0.236 |
| 80-200 m | 0.085 |
| 200-450 m | 0.054 |
| 450-900 m | 0.050 |
| 900-2300 m | 0.022 |

**High Frequency Noise**

It helps to identify critical T drops in the profile (such as large T differences over a large depth) by checking

continual spiking over a wide range of depths (Cowley and Krummel, 2022). In case of continual spikes, values

before and after a chosen interval (4 m approximately, i.e. 7 points) are tested considering the same acceptable

range of T inversion and gradient as in the *inversion and gradient checks* and flagged as bad if they are out of

the ranges.

Figure 2 Example of the QFs generated by the automatic QC tests (Table 2) applied to a temperature profile. The raw profile is at the top left and the final interpolated profile is at the bottom right.

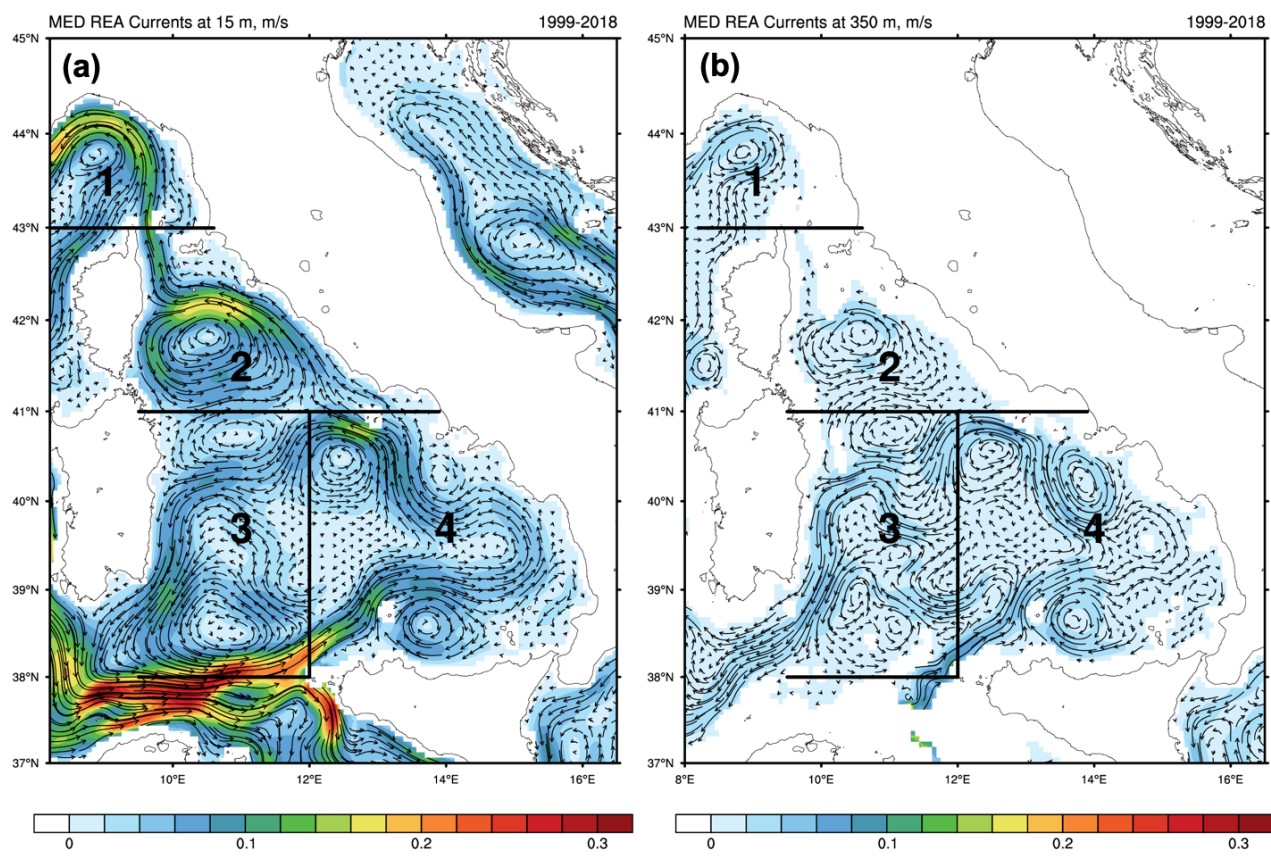

Figure 3 Maps of the mean circulation computed from the Mediterranean Sea reanalysis dataset (Simoncelli et al., 2014) at (a) 15 m and (b) 350 m depth.

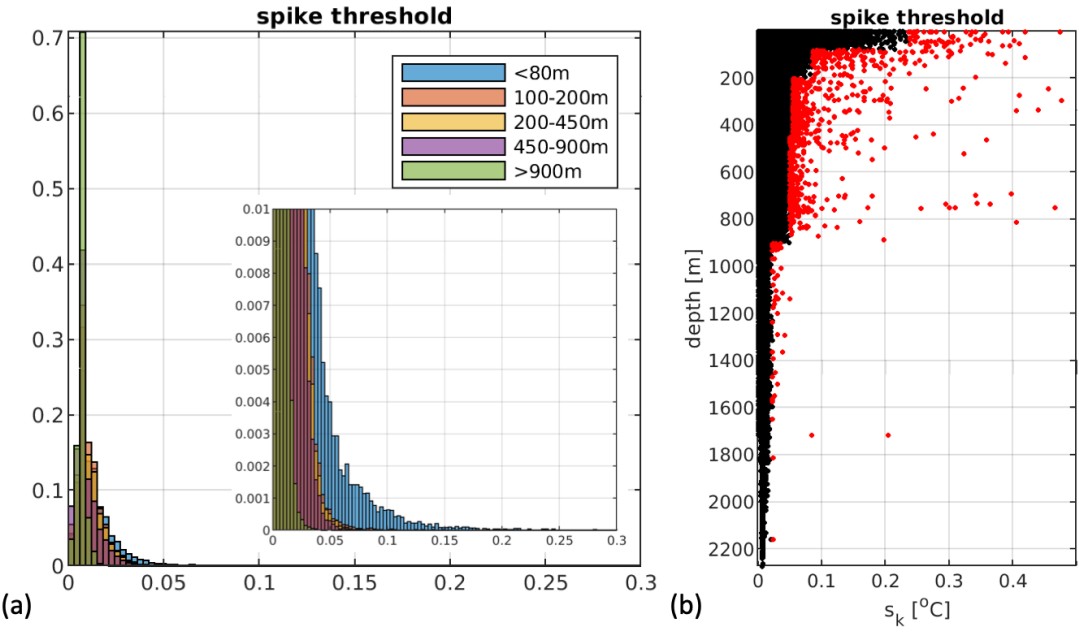

Figure 4 (a) Distribution in terms of probability of the spike threshold ($s_k$) in 5 layers with a zoom probability below 0.1%. (b) Vertical distribution of the spike threshold with indication in red of the values above the 99.99% quantile.

**4.2 Mapping QC test exit values to standard Quality Flags**

Each basic QC test assigns a corresponding exit value to each original depth and T record (Table 2) within the vertical profile in the DEPTH_TEST_QC and TEMPET01_TEST_QC ancillary variables respectively. The mapping of these ancillary variables to QFs is necessary to allow the user to filter the original data according to the quality requirements for the intended use.

The QFs adopted, whose labels and corresponding definition are reported in Table 5, have been selected from the SDN Common Vocabulary (IOC, 2013; IOC, 2019; https://www.seadatanet.org/Standards/Common-Vocabularies). The QF (Table 5) associated with each original T measurement or depth value summarizes the results of the performed automatic tests and it is stored in the dedicated ancillary variable (TEMPET01_FLAGS_QC or DEPTH_FLAGS_QC).

**Table 5 The Quality Flags (QF) selected from the SeaDataNet Common Vocabulary (IOC, 2013; IOC, 2019) assigned to the reprocessed XBT data.**

| id | label | definition |
|---|---|---|
| 1 | good value | Good quality data value that is not part of any identified malfunction and has been verified as consistent with real phenomena during the quality control process |
| 2 | probably good value | Data value that is probably consistent with real phenomena but this is unconfirmed or data value forming part of a malfunction that is considered too small to affect the overall quality of the data object of which it is a part |
| 3 | probably bad value | Data value recognised as unusual during quality control that forms part of a feature that is probably inconsistent with real phenomena |
| 4 | bad value | An obviously erroneous data value |
| 8 | interpolated value | This value has been derived by interpolation from other values in the data object. |

The DEPTH_TEST_QC contains the outcome of two tests, one based on GEBCO local bathymetry (test 2 in Table 2) and one based on the last good depth recorded by the operator (test 3 in Table 2). Since the GEBCO local bathymetry was often in disagreement with the operator information we decided to keep the output of test 3 in DEPTH_FLAGS_QC, considering the operator's annotation more reliable.

The general rule adopted for mapping the QC tests exit values to T QFs is the following:

- GOOD (QF=1) where all the tests pass;
- BAD (QF=4) where at least one of the checks fails.

We decided to use a higher level of detail, introducing also "probably good" (QF=2) and "probably bad" (QF=3) flags, when it's needed, since surface (test 5 in Table 2) and inversion/gradient tests (test 6 in Table 2) can provide more information on profile behavior. After applying general rule for GOOD and BAD flags, we consider the flags coming from the two mentioned tests and we update the flags as follows:

- PROBABLY GOOD (QF=2) if the surface test returns a "probably good" flag;
- PROBABLY BAD (QF=3) if the surface and/or the inversion test returns a "probably bad" flag.

Only measurements that have associated T and depth QFs equal to 1 or 2 have been used for the interpolation at each meter depth. A relative QF associated to the interpolated profile has also been generated in order to

510 label ("interpolated value", QF=8) when there is a gap of more than 5 consecutive points in the original profile,

which coincides with the number of points used to detect spikes (~3 m).

## 4.3 Calibration of the XBT system and correction

As previously highlighted, checking with a tester provides an assessment of the efficiency of an XBT system.

Once a tester is connected to an XBT system in a simulated drop, the tester's measurement indicates how the

XBT system's reading differs from nominal values at some reference temperatures. These differences, which

can be constant or variable over the time interval of data acquisition, can then be used to correct the values of

the XBT profiles. Each tester used during the campaigns on the MX04 line after July 2010 has two reference

temperatures (see Appendix A for details).

Checks, immediately before the first drop and after the last drop, were routinely performed. Further checks

were carried out whenever the computer or DAQ had failures. The differences measured at the reference

temperatures at the start/end of each MX04 cruise are shown in Figure 5a, while their drift during a cruise is

shown in Figure 5b. The values vary marginally and slightly over the time, but large anomalies occurred in

September 2013 (cruise 14) and June 2014 (cruise 18) for unknown reasons. The DAQ used in those campaigns

showed an initial offset followed by a random and oscillating variability throughout the day: for example, the

recorded values during the checks in June 2014 were 26.678 °C (start), 26.649 °C, 26.668 °C and 26.666 °C

(end) instead of 26.758 °C. This type of anomaly was also found from Reseghetti et al. (2018) during XBT vs.

CTD comparison tests, where it was pointed out that the T differences between the XBT and CTD profiles

were heavily affected by the DAQ functioning.

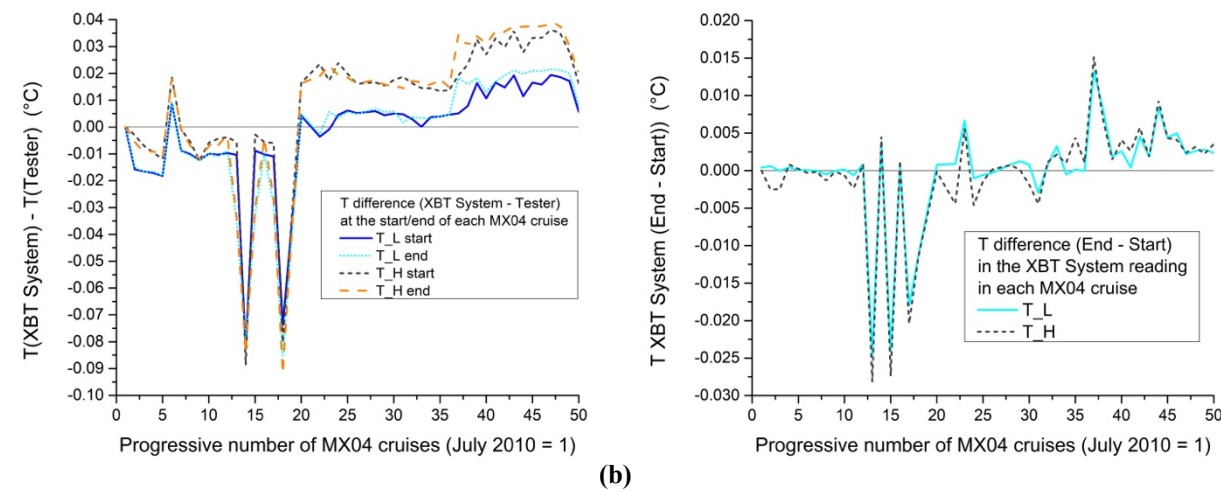

**(a)**                                                                                          **(b)**

**Figure 5 (a) Temperature difference (XBT System-Tester) obtained from the checks at the reference temperatures before starting and at the end of each MX04 cruise. (b) Difference between initial and final measurement with the tester during the same cruise at the reference temperatures.**

### 4.3.1 Correction Algorithm

The measurements with a tester are used to correct the T values of each XBT profile of a campaign under the assumption that the difference between the initial and final tester readings at reference temperatures varies linearly over time from the beginning to the end of the campaign. The reference values are obtained by calculating the average resistance value over the last 30 consecutive recorded values at each temperature in the simulated drop (i.e. 3 seconds of acquisition, with a sampling frequency of 10 Hz) and then converted into T values (for details, see Appendix A). The differences between the nominal temperatures and the read values are linearly interpolated as a function of the time elapsed since the first launch to calculate their hypothetical value in correspondence with each XBT probe during the campaign. In case of a single-point tester, a constant correction is added to each value of the XBT profile. In case of two-point tester, the correction is obtained by a further linear interpolation, based on the differences at upper and lower temperatures of this tester.

Notation:

- N is the number of XBT probes deployed during the campaign;
- $T_+$ and $T_-$ nominal upper and lower temperature on the tester;
- $\Delta T_{+,i}$, $\Delta T_{+,f}$ initial and final temperature difference at the value $T_+$ ;
- $\Delta T_{-,i}$, $\Delta T_{-,f}$ initial and final temperature difference at the value $T_-$ ;
- $t_i$, $t_f$ initial and final time of the XBT drops (usually, $t_i$ is set to 0);
- $t_k$ time elapsed from the initial check with the tester, which is assumed to be coincident with the first XBT drop ($1 \leq k \leq N$);
- $T_{+,k}$ and $T_{-,k}$ theoretical upper and lower temperature that the tester should read at the k-th drop.

These last values can be calculated as

$$T_{+,k} = T_{+,i} + \Delta T_{+,k} \quad \text{and} \quad T_{-,k} = T_{-,i} + \Delta T_{-,k}$$

where the estimated difference at upper and lower reference T corresponding at the *k* drop are:

$$\Delta T_{+,k} = -\left[\Delta T_{+,i} + \left(\frac{\Delta T_{+,f} - \Delta T_{+,i}}{t_f - t_i}\right)(t_k - t_i)\right] \text{ and } \Delta T_{-,k} = -\left[\Delta T_{-,i} + \left(\frac{\Delta T_{-,f} - \Delta T_{-,i}}{t_f - t_i}\right)(t_k - t_i)\right]$$

The so calculated contributions are combined in the correction term for the specific *k* XBT:

$$\Delta T_{corr,k} = \left(\frac{\Delta T_{+,k} - \Delta T_{-,k}}{T_+ - T_-}\right)\left(T_{read,k} - T_-\right) + \Delta T_{-,k}$$

and then added the original value $T_{read,k}$ recorded by the DAQ:

$$T_{corr,k} = T_{read,k} + \Delta T_{corr,k}$$

$T_{corr,k}$ is thus the value that best represents the actual seawater temperature measured by the *k* XBT probe assuming that the calculated correction (based on the initial and final measurements provided by the tester) is the best way to describe how the XBT system operates when the probe was deployed. Obviously, $\Delta T_{corr,k}$ is not related to the measurement quality due to the probe characteristics or to possible issues during data acquisition. When the calibration is available, the correction calculated in this way has been applied to the raw data prior to the QC analysis but it is also provided as a separate variable (CALIB) so that the user might decide to remove it. This correction must absolutely not be applied to the profiles from XCTD-1 probes because their

acquisition circuit works in a completely different way and the shipboard DAQ simply acts as a data receiver

and does not play an active role in the measurement.

## 4.4 Vertical interpolation

Three interpolation methods were tested: linear (LI), RR (Reiniger and Ross, 1968) and MR-PCHIP (Barker

and McDougall, 2020). The goal is to select the most conservative method, i.e. the one that provides the closest

interpolated T values to the original reading. The original measurements of each XBT profile were subsampled,

discarding half of the measurements then used as control values against the newly interpolated ones to calculate

differences and Root Mean Square Differences (RMSD) and therefore evaluate the best interpolation method

for our dataset.

Original values have been interpolated with the three methods on the control depth levels and the resulting T

estimates have been compared with the measured ones. Figure 6 shows an example of an observed profile with

highlighted control levels (magenta), the interpolated profile with the three considered methods and the relative

differences (interpolated-original). Figure 6a presents an example of the large T differences that occur between

interpolated and measured values (0.4 ºC or -0.2 ºC) along the thermocline at about 35 m. Figure 6b shows a

step-like profile below 600 m depth where the differences are very small, less than 0.02 ºC, but they can

slightly increase and differ among the three methods where T vertical gradients occur.

Mean bias and RMSD have been computed in vertical bins (766) of 3 m thickness and the obtained metrics

profiles are displayed in Figure 7, associated with their relative vertical data distributions. These metrics have

been computed for the whole dataset and for two separate time periods: from June to November (when the

thermocline is well developed) and from December to May (when the water column is more homogeneous).

The mean bias in Figure 7 presents values in the range (-0.001, +0.001) ºC, the interval halves from December

to May whereas it practically doubles (-0.002, +0.001) ºC from June to November. The maximum RMSD

when considering all profiles is about 0.04 ºC, it halves from December to May while it is close to 0.06 ºC

from June to November. Except for the Dec-May plot, the maximum RMSD values are associated with LI and

RR methods but we note that RMSD < 0.01 ºC for the three methods below 100 m depth.

The total RMSD on the entire water column has been summarized in Table 6 for the three time periods and

the surface layer above 100 m. In fact, the total bias estimated is zero for the three methods and the three time

periods, while the total RMSD is 0.011 ºC for LI, 0.011 ºC for RR and 0.010 ºC for MR-PCHIP, while in the

surface layer the values are 0.023 °C, 0.021 ºC and 0.019 °C respectively. The maximum RMSD values usually

occur during the stratified period (Jun-Nov) with values equal to 0.013 ºC for LI, 0.012 ºC for RR and 0.011

599     ºC for MR-PCHIP, that in the surface layer become 0.030 °C, 0.027 °C and 0.023 °C, respectively.

The computed metrics in vertical bins present very small values, much lower than and the specified T

uncertainty (0.10 °C). However, the absolute differences in the surface layer when the thermocline settles can

be larger than 0.2 ºC as in Figure 6. The MR-PCHIP interpolation always presents the smallest error for the

analyzed dataset (Table 6) with respect to the reference values, thus it has been applied.

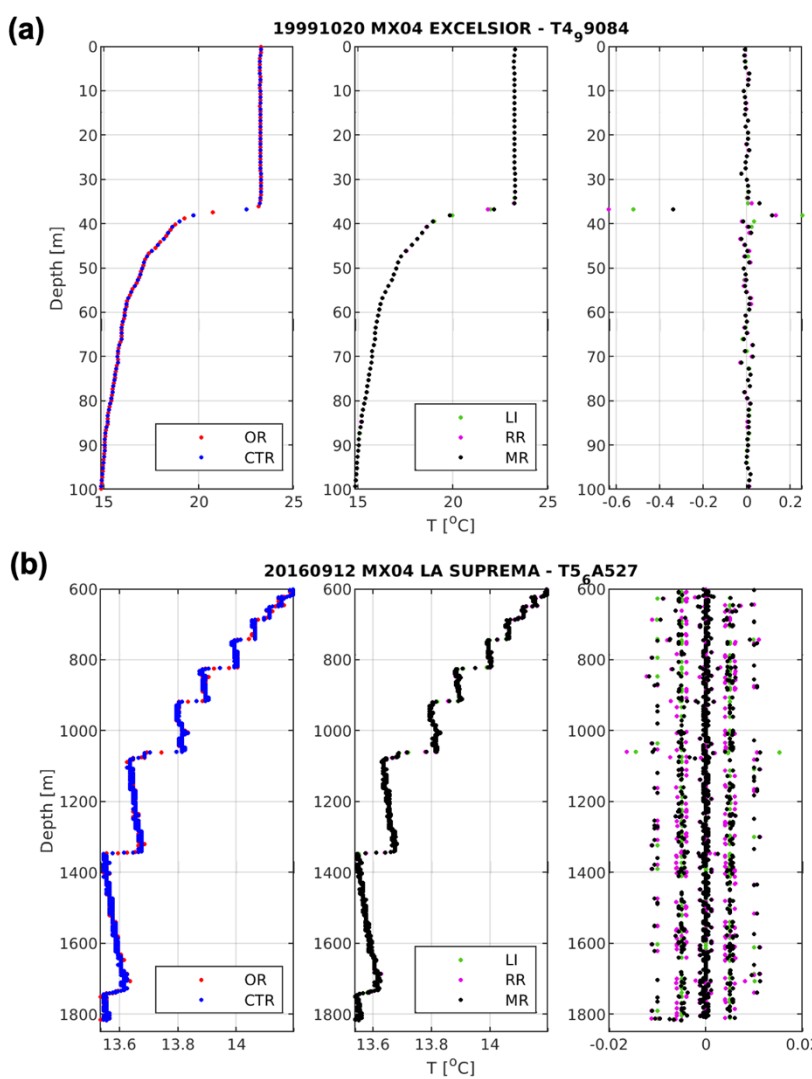

**Figure 6 Temperature profiles in the surface layer 1-100 m (a) and in the deep layer 600-1800 m (b): (left) magenta dots represent the control records; (middle) interpolated temperature values with linear LI (linear) , RR (Reiniger and Ross, 1968) and MR-PCHIP (Barker and McDougall, 2020); (right) differences between the interpolated and measured T values.**

**Table 6 Summary of the computed metrics from the three interpolation methods: linear (LI), RR and MR-PCHIP Temperature RMSD [°C] have been computed in the entire water column and in the surface layer (0-100 m) from the whole dataset (All) and in two time periods December-May (mixed) and June-November (stratified).**

| RMSD | LI | RR | MR-PCHIP |
|---|---|---|---|
| **All** | 0.011 | 0.011 | 0.010 |
| **0-100 m** | 0.023 | 0.021 | 0.019 |
| **Dec-May** | 0.010 | 0.010 | 0.010 |
| **0-100 m** | 0.014 | 0.014 | 0.013 |
| **Jun-Nov** | 0.013 | 0.012 | 0.011 |
| **0-100 m** | 0.030 | 0.027 | 0.023 |

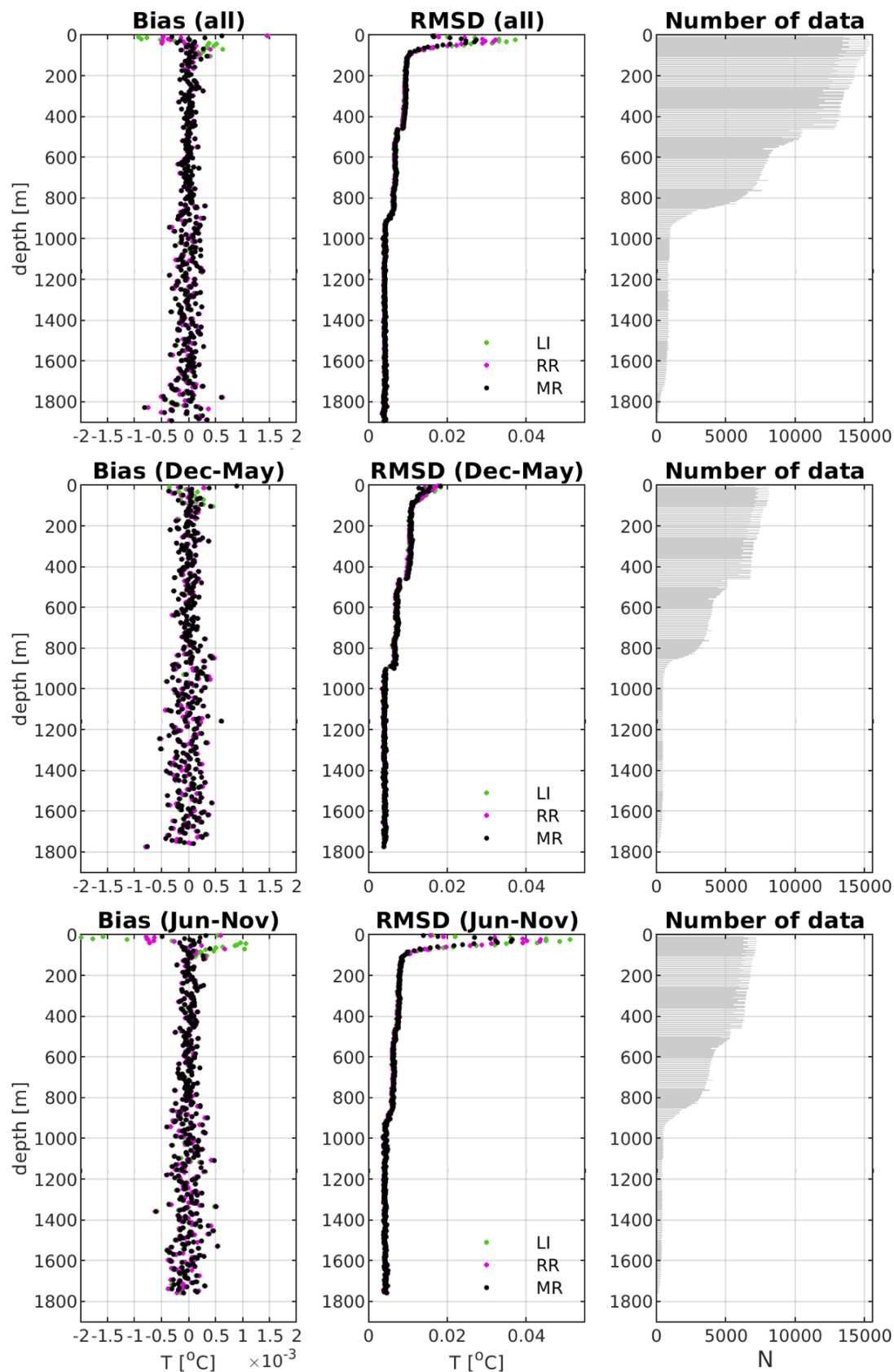

**Figure 7 Profile of mean bias (left) and RMSD (middle) computed from profiles interpolated on selected depths and compared to the corresponding measured values considering the three methods: linear (LI), MR-PCHIP (MR) and Reniger and Ross (RR). Three different time spans are shown: (top) the whole dataset; (middle) from December to May; (bottom) from June to November. (right) Vertical data distribution in 3 m bins.**

## 5. Results

The QC algorithms applied to the dataset are not capable of catching all erroneous values. According to Good et al. (2023) any automatic QC test produces a percentage of True Positives (TP, correctly detected erroneous data) and False Positives (FP, incorrectly detected erroneous data) and the general aim would be to maximize the TP (correct flagging) rate and minimize the FP (incorrect flagging) rate.

The new automatic QC procedure has been tuned using visual checks to reach an optimal TP/FP rate. Specifically, efforts have been made to tune the vertical gradient and spike thresholds, using quantiles analysis to maximize the detection of erroneous data (TP) and minimize flagging of GOOD data as BAD (FP). This was particularly tricky for the vertical gradient test which detected 121 profiles with out of bounds values, but 28 of them appeared FPs (FP/TP rate of 23%) from visual check. In fact, the strong seasonal stratification of the Mediterranean Sea and the presence of several water masses in different water layers might cause the incorrect flagging of GOOD data as BAD (FP), as shown in Figure 8b,d. This makes the vertical gradient test non-optimal for the Mediterranean Basin with a high FP rate, thus a very small percentage associated with the quantiles have been selected to minimize this.

The spikes test is much more effective (331 profiles with detected spikes of which 11 are FPs), providing a low FP/TP rate (3.3%). Figure 9 shows example profiles with TP spikes (a) and FP spikes (b), mainly marked at the start of the thermocline.

However, some profiles present anomalous features that automatic QC procedure could not detect. The decision was to add a flag associated with the whole profile indicating the depth range where unrecoverable problems began. The decision is based on the knowledge of the main physical characteristics of the water masses present in the analyzed region. In fact, the very small Rossby radius (~11 km on average) and the occurrence of repeated and well-documented thermal inversions must always be considered when the quality of the T profiles is analyzed. Step-like structures ("staircases") are also typical of the southern Tyrrhenian Sea, explained usually in terms of the double diffusion process (Meccia et al. 2016; Durante et al., 2021).

Sometimes, the meteorological conditions and a non-accurate knowledge of the bathymetry can make the expert validation of XBT profiles difficult, but their extreme variability can also be ascribed to multiple instrumental and operational factors. In every XBT drop, the correct unwinding of the wire from both spools, adequate and complete protection of the insulating substance along its entire length are essential to guarantee good quality of the recorded data. For example, most profiles from XBTs launched from ships traveling at low speed (i.e. v < 15 knots, less than 10% of the dataset) are generally less affected by significant electrical disturbances, even in the presence of wind. Unfortunately, the ships used on the MX04 line (from which most of the REP profiles belong) have a standard speed close to 22 knots and this makes the acquisition conditions vulnerable. The XBT profiles from containerships also have a lower quality due to the usually very high launch position (h > 25 m), which makes the probe depth in the initial measurements provided by software questionable (Bringas and Goni, 2015). As mentioned in section 2, the electric current that circulates in the unwinding copper wire transforms it into an antenna sensitive to all electromagnetic phenomena occurring in nearby. The occurrence of atmospheric events (thunderstorms with lightning) can have a non-negligible impact

on the recorded signal, same as the proximity to on-board instrumentation producing significant electromagnetic fields and whose operation is random. The physical parameter measured by the XBT system is the electrical resistance, which has two components: one is from the copper wire and the other from the NTC thermistor which falls through the water column. Gusts of wind combined with turbulence produced by the ship hull can produce "whiplash" on the copper wire and badly influence the shape of the profiles collected with particularly unfavorable wind conditions.

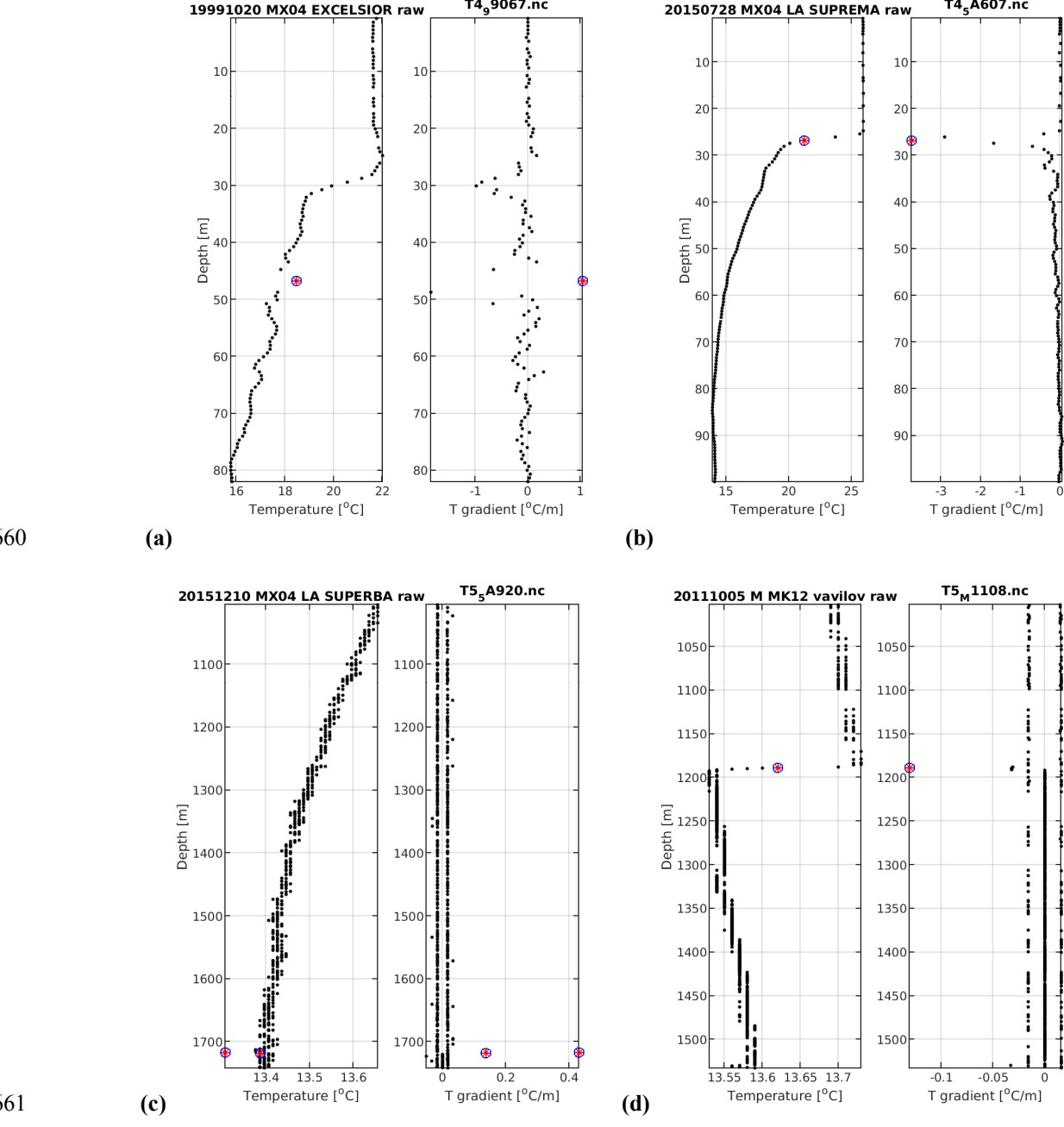

**Figure 8 Examples of temperature gradient flags applied to different XBT profiles: (a) true positive vertical gradient anomaly in the surface layer; (b) false positive vertical gradient anomaly in the surface layer; (c) true positive vertical gradient anomaly in the bottom layer; (d) false positive vertical gradient anomaly in the bottom layer. The sub-plots have different axes ranges.**

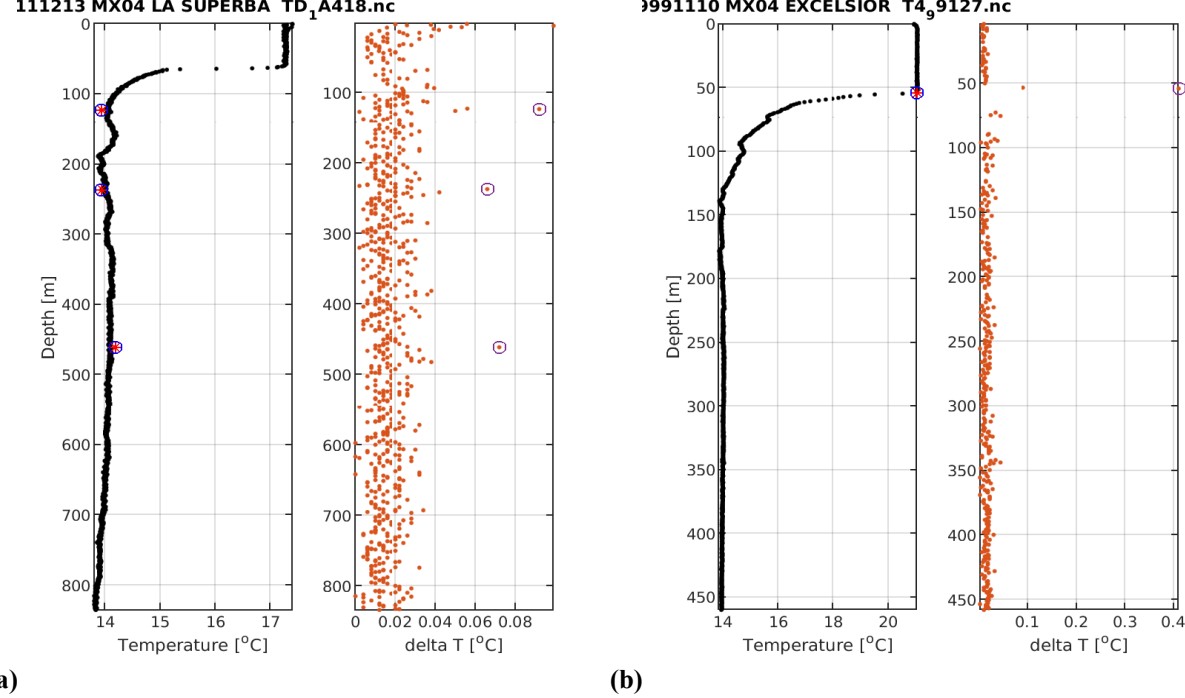

**(a)**                                                                                         **(b)**

**Figure 9 Examples of spikes detected in two different XBT profiles: (a) true positive spikes; (b) false positive spike at the start of a steep thermocline. The orange dots in the right panels of (a) and (b) indicate the estimated value of the $s_k$ parameter having $c_k$ not equal to zero. The sub-plots have different axes ranges.**

A difficult task has been how to identify the external influences that cause high frequency noise in the T profile, as in the examples of Figure 10 c-d-e, and how to annotate it in the metadata. Some other anomalous thermal structures, compared to what is expected in a certain period, region and depth layer are shown in Figure 10 a-b and f. The visual check carried out by the expert allows in some cases to highlight notable deviations in the shape and/or values of a profile compared to adjacent ones. The probability of having the same type of anomalous structure recorded by two adjacent XBT probes in time and space is considered negligible, favoring the occurrence of something physical instead of non-optimal functioning of a specific probe. Sometimes the initial BAD attribution to anomalous structures was subsequently reviewed by the comparison with adjacent profiles that present similar features (e.g. Fig.10 a).

## 5.1 Comparison with SeaDataNet data version

A significant part of the XBT profiles included in this dataset have been systematically disseminated through the SDN infrastructure and can be accessed from the data access portal through the saved query Url https://cdi.seadatanet.org/search/welcome.php?query=1866&query_code={4E510DE6-CB22-47D5-B221-7275100CAB7F}). Alternatively, they can be found in the Mediterranean aggregated dataset product (Simoncelli et al., 2020a) in which they are integrated with other data types (CTDs, bottles, MBTs, profiling floats). This data product has been further validated in the framework of the SeaDataCloud project (https://www.seadatanet.org/About-us/SeaDataCloud), as described in Simoncelli et al. (2020b).

The SDN XBT dataset, extracted from Simoncelli et al. (2020a) is considered here as a benchmark to highlight the main effects of the proposed data reprocessing. Bias and RMSD profiles have been computed from 3104 matching profiles with a vertical data distribution shown in Figure 11. Since SDN profiles do not have the calibration correction, we have computed the separate metrics with and without the correction applied. The black dots represent all matching profiles, green dots represent the profiles without correction and the red dots have the correction applied.

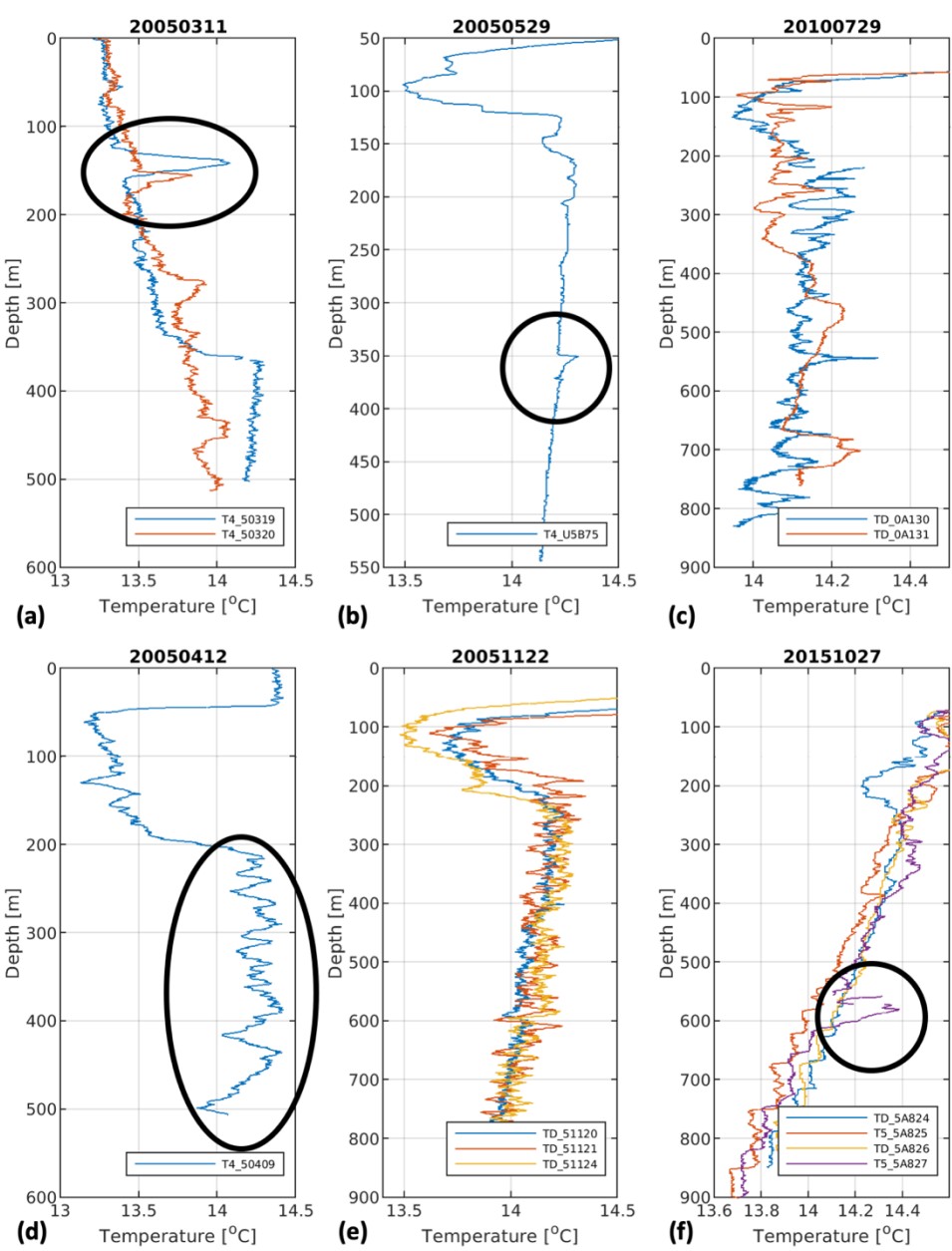

**Figure 10 Examples of profiles with critical features: (a-b-f) anomalous thermal structures; (c-d-e) profiles affected by high frequency noise. The name of the selected profiles is shown in the legend. The sub-plots have different axes ranges.**

The maximum discrepancy among the two data versions resides always within the surface layer until 150 m depth. The maximum bias and RMSD reach approximately 0.05 °C and 0.2 °C respectively, which might imply

potential significant changes in downstream applications. The bias is larger (~0.06 ºC) when estimated from
profiles without calibration correction and slightly smaller (~0.04 ºC) from calibrated profiles, while the largest
RMSD derives from profiles with the correction applied, indicating that the correction slightly increases on
average the REP temperature values and consequently the positive bias.

The REP profiles are warmer than SDN ones in the surface layer and below 900 m, while between 150 m and
800 m both metrics are small and consistent. The overall mean bias and RMSD are equal to 0.002 ºC and 0.041
705 ºC, respectively. Such differences are mainly due to the new interpolation technique, the lack of filtering, the
706 application of the calibration correction in the REP dataset, and in very few cases, the use in SDN of wrong
FRE coefficients or the incorrect probe type assignment which can produce a change of the depth values. The
sharp reduction in the number of observations available below about 900 m depth and the application of the
tester correction affect the shape of both BIAS and RMSD profiles.

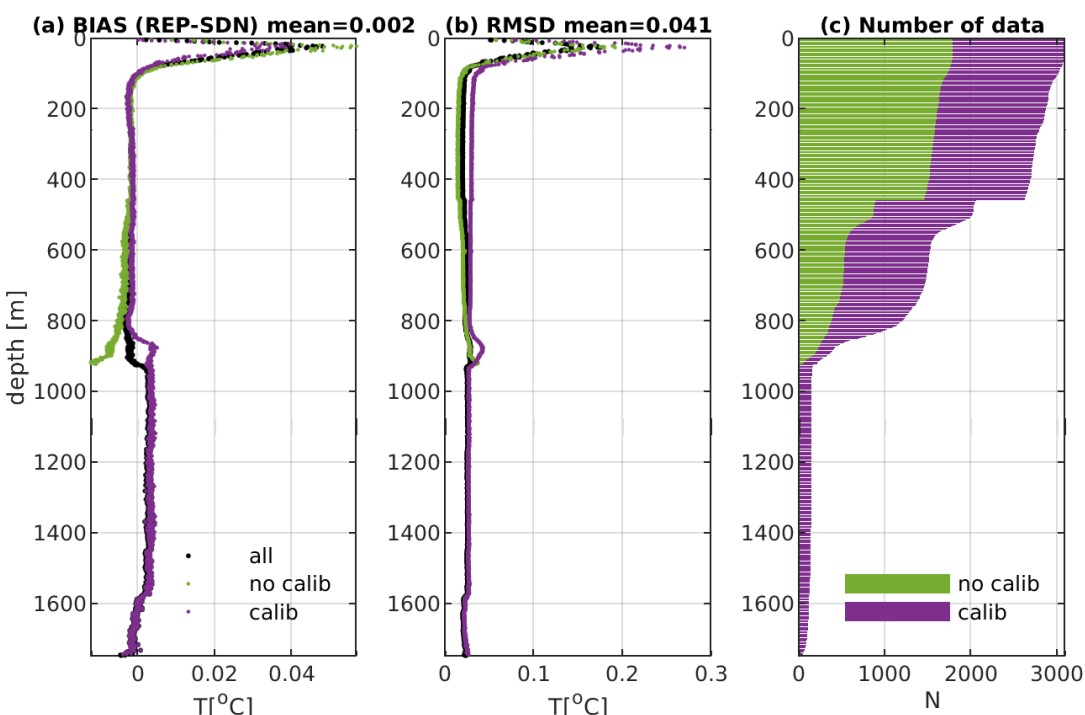

**Figure 11 Comparison between the reprocessed (REP) and the corresponding SeaDataNet (SDN) profiles at each**
**meter depth: (a) Bias mean profile; (b) RMSD profile and (c) cumulative vertical data distribution which shows**
**the relative contribution of profiles with calibration and profiles without calibration to the total.**

Figure 12 shows examples of matching REP and SDN profiles and their difference with a zoom in the surface
(a) and bottom layer (b and c), where the largest differences occur. During the stratified period, the largest
differences reside in the thermocline and can exceed 1.5 °C (Figure 12a), while in the bottom layer the
calibration correction (see Figure 12b, c) together with the abrupt decrease of the number of data explain the
small positive average bias in Figure 11a. In fact, numerous T5/20 profiles (maximum rated depth, see Table
1) were launched (~7% of the total) in the few campaigns in which the acquisition system showed significant
negative anomalies and this influenced both BIAS and RMSD profiles below 900 m depth. The frequent step-

like shape of deep profiles (Figure 12c), due to double diffusion processes (Meccia et al. 2016; Durante at al., 2021), causes instead positive spikes in the difference profiles.

In the SDN dataset, the interpolation of raw profiles at each meter depth has been combined with the application of a Gaussian filter to reduce possible noise (Manzella et al., 2003 and 2007). Consequently, a general smoothing of T profiles is observed, which is appreciable to remove/reduce unrealistic high frequency oscillations, if needed, but it also affects the values of the whole profile. The main effect is that the shape of thermal structures is smoothed out, more or less evidently depending on the recorded T gradient.

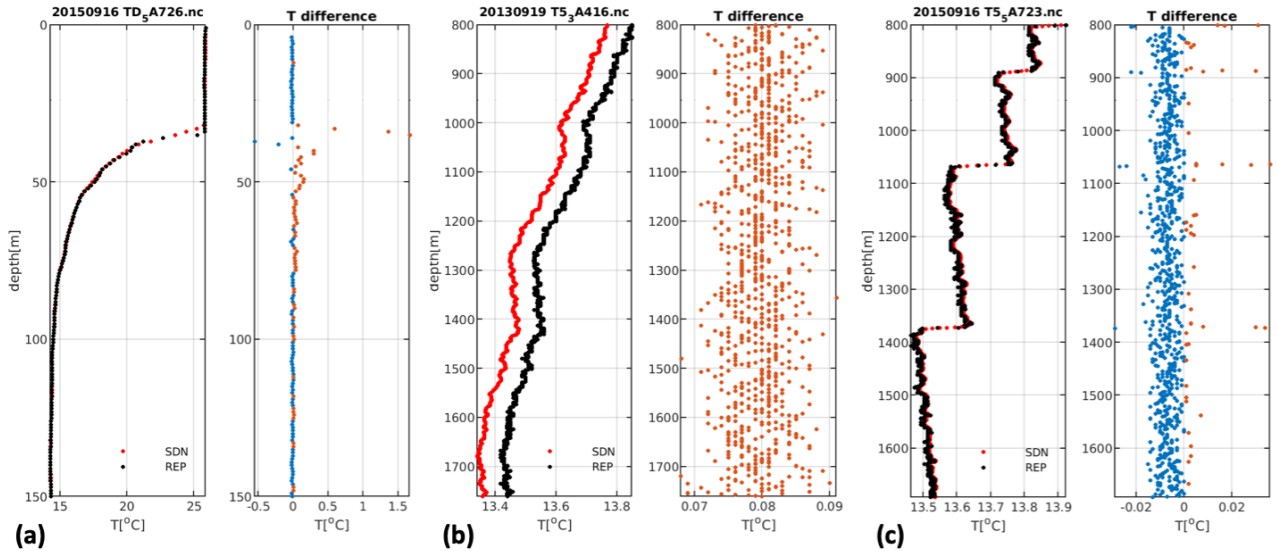

**Figure 12 Example of a reprocessed (REP) profile and the corresponding SeaDataNet (SDN) one on the left and their difference on the right: (a) zoom in the surface layer 0-150 m; (b and c) zoom in the bottom layer below 800 m.**

## 6. Summary and Conclusions

This work presents the reprocessing of XBT profiles in the Ligurian and Tyrrhenian Seas over the time period 1999-2019. The added value of this analysis is the availability of the original raw data and all the metadata from the operational manual notes. This allowed us to create the most complete dataset possible with metadata accompanying each individual T profile. The surface measurements have been added with quality indication and a correction from calibration has been applied, when available, to T values (generally in the range 0.01-0.02 °C), representing the best estimate of the thermal offset due to the operating XBT system characteristics. A new automatic QC procedure and a new vertical interpolation (Barker and McDougall, 2020) have been implemented without the application of any filter that: on one side, removes unrealistic high frequency oscillations, and on the other, it smooths out the thermal structure of the T profiles with main impact on the surface layer during stratified conditions. The adoption of a Gaussian filter in SDN data (Manzella et al., 2003; 2007) was justified by the purpose of assimilating XBT profiles in the Mediterranean Forecasting System that in the early 2000s was characterized by a much lower resolution compared to the present numerical model capabilities. Cheng et al. (2014) XBT bias correction scheme for both temperature and depth records has also been applied to the calibrated profiles, in agreement with the recent literature, to facilitate the REP dataset

integration with other data types for climate studies. The REP dataset gives researchers the most complete information for its re-use for different applications (assimilation in ocean and climate models, process and climate studies). It can also be used to test new QC algorithms or the order on which to apply them to further improve the data quality.

The adoption of FAIR data management principles through the use of SeaDataNet standards and the dissemination strategy based on the ERDDAP server implementation are additional values of this effort, allowing its machine to machine access.

XBTs are a 60-year-old technology. Though the quality of their measurements might not fit the purpose of all applications and they leave debris in the ocean, "XBTs provide the simplest and most cost-efficient solution for frequently obtaining temperature profiles along fixed transects of the upper ocean" (Parks et al., 2022) using ships of opportunity. Moreover, the XBT measurements along the MX04 track were for some periods among the few measurements recorded in the Tyrrhenian and Ligurian Seas. Despite the limitations of the XBT characteristics, they constituted the simplest way to verify the physical state of the upper layer of those basins. It is therefore very important to provide those profiles with the best quality and usability indications. For this reason, the MX04 line has been re-established on a seasonal base in the framework of the MACMAP project after a two-year break for climate monitoring.

In recent years, the use of XBTs has also been criticized because all probe components fall to the seabed. Given the current MACMAP sampling strategy with 37 launches in fixed and determined positions along the MX04 line, the quantity of material abandoned at sea for each campaign can be easily estimated (about 22 kg of ZAMAK, just over 2 kg of plastic and about 11 kg of copper wire). It would be preferably that the XBT probes were made of alternative materials (e.g, iron "nose" and biodegradable plastic components), however, in our cost-benefit analysis, the environmental impact due to the REP dataset is balanced by the scientific results. Finally, the deployment of the XBT probes described here did not contribute to additional emissions of $CO_2$ and other atmospheric pollutants, because only ships of opportunity were used and in the case of research vessels, the launch of the XBT probes was ancillary to the primary purpose of the scientific cruise.

## 7. Data Availability and FAIRness

The management of the REP dataset has been conceived since the beginning to be compliant with the FAIR data management principles (Wilkinson et al., 2016) and the open science paradigm. The REP dataset (Reseghetti et al., 2024; https://doi.org/10.13127/rep_xbt_1999_2019.2) is available and accessible through INGV (Bologna) ERDDAP server (http://oceano.bo.ingv.it/erddap/index.html), which allows machine to machine data access, enables downloading subsets of the dataset and gives to the users the possibility to select among several download formats. ERDDAP is a FAIR-compliant data access service (O'Brien and Delaney, 2024) in line with the GOOS (Global Ocean Observing System) Observations Coordination Group (https://goosocean.org/who-we-are/observations-coordination-group/) strategy. In fact, according to Lange et al. (2023), ERDDAP "(i) supports dozens of popular formats; (ii) provides standards-based metadata and data services and formats; (iii) supports federated access of distributed ERDDAP data services; (iv) supports both

human and machine interactions; (v) supports sub-setting of large datasets; (vi) provides improved discovery of datasets through commercial search engines; and (vii) provides support for archival of datasets". The REP dataset is machine-readable, enabling its automated transfer, through a federated ERDDAP server's approach, to other repositories and marine data infrastructures, such as EMODnet Physics (https://emodnet.ec.europa.eu/en/physics) (Novellino et al., 2024).

The raw data with calibration information, bias correction and the interpolated data at standard depths after data QC are released with complete metadata description together with all the processing information in order to facilitate data reuse. The metadata are available through *url_metadata* variable (Appendix C.6). Data and metadata of each profile can be easily associated through the *profile_id* and *cruise_id* fields. To facilitate data reusability, we prepared a Jupyter Notebook in Python that allows recombining all data and metadata in NetCDF files, one per XBT profile. The notebook (Fratianni and Frizzera, 2024) is available on a GitHub repository and published on Zenodo.

The standards adopted for the dissemination of the REP dataset are described in detail in Appendix C.

The ODV collection of the REP interpolated dataset, used for the visual check, is also available on request.

**Author contribution**

SS conceptualized the work, FR curated the original data (collecting a significant portion of it), CF developed the QC software, under the methodology supervision of SS, FR and LC. GR prepared the correction from the calibration of DAQs. CF manages and curates the reprocessed dataset. SS, FR and CF prepared the manuscript with contributions from GR and LC.

**Competing interests**

S. Simoncelli is a member of the editorial board of the journal. Co-authors declare that they have no conflict of interest.

**Acknowledgements**

We thank all people/institutions/companies involved in the data taking:
- The Italian shipping company GNV, a very special partner that has allowed the monitoring activity since September 1999: in particular Marco Fasciolo, Dr. Mattia Canevari, the captains, the officers and all the crews for their precious collaboration;
- Persons involved in data collection on the MX04 line, namely M. Borghini, F. Dell'Amico, C.Galli, E. Lazzoni (CNR-ISMAR), M. Morgigni and A. Baldi (ENEA-STE);
- CNR-ISMAR-Lerici for the very long collaboration that has allowed the acquisition of numerous XBT profiles from research vessels, in particular the crew and technicians of the RV Urania;

●    The international shipping companies Hapag Lloyd, CMA CGM and Arkas, their managers and crews
for their valuable collaboration;

●    Responsible officers ashore and on board, crews and technicians of ships belonging to IIM, in
particular CF Maurizio Demarte and Dr. Luca Repetti.

●    Australian government agency CSIRO for its kind cooperation by sharing their instrumentation in the
2007-2011 data collection on container ships, notably Dr. Ann Thresher, Dr. Lisa Krummel and
Rebecca Cowley;

●    The Federal Research Laboratory NOAA-AOML of Miami (FL), in particular Dr. Gustavo Goni and
Dr. Francis Bringas, for the supply of the XBT probes used during some MX04 campaigns and for the
support in carrying out the operational activities;

●    Stefano Latorre (INFN, Milan), key person in the development and implementation of the testers and
their periodic calibration;

●    One of the authors (FR) for having supplied his own instrumentation and XBT probes for carrying out
oceanographic campaigns since 2008.

A very special thanks to Giuseppe M. Manzella, who created the SOOP program in the Mediterranean Sea and
coordinated it until 2013 and was among the pioneers in the development of marine data infrastructures. He
supported this paper, providing useful comments.

We acknowledge Marjahn Finlayson for reviewing the English, and Mario Locati (head of the INGV data
management office) for his continuous support. This work has been developed in the framework of the
MACMAP project, funded by Istituto Nazionale di Geofisica e Vulcanologia (Environment Department), and
coordinated by Antonio Guarnieri that we thank.

# Appendix A

**Characteristics of test canisters**

While in the laboratory, it is easy to have steady and controlled environmental conditions for measurements, in the field, this is only an aspiration of the operators. Furthermore, repeated operation in conditions of high temperature, humidity and salinity certainly does not facilitate the proper functioning of the electronic instrumentation. The DAQ in an XBT system should read the nominal value of a resistance (within the uncertainties of the measurements) showing no changes in its reading over time. The use of a tester with high quality resistors is the preferred method to verify this. Between 2007 and 2010, two testers were built using very high precision resistors (model KOA-Speer RN73r1jttd1002b10) combined in such a way as to achieve corresponding T values similar to the extreme ones measured in the marine regions under investigation. The resistance values of both testers were checked each year with a Wavetek Datron 1281 8.5 digits multi-meter in a laboratory of the INFN (Italian National Institute of Nuclear Physics) in Milan (room temperature always in the range 20-24 °C during measurements). The reading remained stable (within 0.1 Ohm) over the period 2008-2019 for the former and 2010-2015 for the latter.

**Table A1 - The resistance values measured in the control tests with the corresponding temperature values calculated by a Hoge_2 equation for the two testers used in the XBT data acquisition campaigns since 2010.**

| Model | Resistance 1 (Ohm) | Temperature 1 (°C) | Resistance 2 (Ohm) | Temperature 2 (°C) |
|---|---|---|---|---|
| Test canister 1 | 4631.0 ± 0.1 | 26.758 ± 0.001 | 8960.1 ± 0.1 | 12.197 ± 0.001 |
| Test canister 2 | 4397.2 ± 0.1 | 27.956 ± 0.001 | 8725.3 ± 0.1 | 12.759 ± 0.001 |

The resistance R values shown in Table A1 are then converted to T by applying the Hoge_2 R to T equation (Sippican, 1991 and 2010; Hoge, 1988; Chen, 2009; Liu et al., 2018)

$$T = \frac{1}{A + B(\ln R) + C(\ln R)^2 + D(\ln R)^3} - 273.15°C$$

with the following coefficients: A = $1.2901230 \cdot 10^{-3}$, B = $2.3322529 \cdot 10^{-4}$, C = $4.5791293 \cdot 10^{-7}$, D = $7.1625593 \cdot 10^{-8}$

To our knowledge, this equation and the coefficients remained unchanged since the 1990s for all the DAQs, , namely Sippican MK12, MK21 ISA, MK21 USB, MK21 Ethernet, Turo Devil, Turo Quoll. Sippican used the Steinhart-Hart relation for its MK9 model (IOC, 1992) while tabulated R to T values were used for MK-2A and similar recorders (Sippican, 1968; Plessey, 1975).

# Appendix B

**Table B1 - Summary of ships, instrumentation and operating conditions during the collection of the XBT profiles in the REP dataset.**

| Ship Name | Call Sign/ IMO No. | Number of Campaigns | Years of Activity | DAQ used | Height launch (m) | Range of ship speed (knots) |
|---|---|---|---|---|---|---|
| **Excelsior** | IBEX 9184419 | 20 1 7 | 1999-2000 2012 2017-2018 | MK12 MK21 USB MK21 Ethernet | 10±0.5 | 20-24 |
| **Excellent** | IBBE 9143441 | 1 5 | 2004 2012-2014 | MK21 ISA MK21 USB | 10±0.5 | 19-24 |
| **Splendid** | IBAS 9015747 | 1 | 2011 | MK21 USB | 10±0.5 | 20-22 |
| **La Superba** | ICGK 9214276 | 14 1 23 1 3 | 2004-2006 2010 2010-2016 2011 2016-2017 | MK21 ISA TURO QUOLL MK21 USB MK12 MK21 Ethernet | 11±0.5 | 21-28 |
| **La Suprema** | IBIL 9214288 | 2 6 6 | 2004 2011-2016 2016-2019 | MK21 ISA MK21 USB MK21 Ethernet | 11±0.5 | 21-28 |
| **Wellington Express** | DFCX2 9224051 | 5 | 2007-2008 | TURO DEVIL | 25±1.0 | 14-20 |
| **Canberra Express** | DFCW2 9224049 | 1 | 2008 | TURO DEVIL | 25±1.0 | 14-20 |
| **Stadt Weimar** | DCHO 9320051 | 8 | 2009-2010 | TURO DEVIL | 27±1.0 | 14-20 |
| **CMA CGM Charcot** | A8HE4 9232773 | 5 | 2009-2011 | TURO DEVIL | 25±1.0 | 14-20 |
| **Daniel A** | TCLA 9238064 | 2 | 2014 | MK21 USB | 8±0.5 | 14-17 |
| **Ammiraglio Magnaghi** | IGMA 8642751 | 3 1 2 | 2008-2013 2011 2019 | MK12 MK21 USB TURO QUOLL | (3 − 6) ±0.5 | 1-10 |
| **Aretusa** | IABA | 1 2 | 2006 2017-2018 | MK12 MK21 USB | (4 − 5) ±0.5 | 1-10 |
| **Galatea** | IABC | 1 | 2013 | MK12 | (4 − 5) ±0.5 | 1-10 |
| **Urania** | IQSU 9013220 | 12 13 | 2000-2012 2005-2014 | MK12 MK21 USB | (3 − 12) ±0.5 | 0-11 |
| **Minerva 1** | IZVM 9262077 | 1 1 | 2015 2016 | MK21 USB MK21 Ethernet | (3 − 8) ±0.5 | 0-11 |
| **Ibis** | -- | 1 | 2019 | MK21 Ethernet | 3 ±0.5 | 0-10 |

## Appendix C

**Format and standards**

The data format adopted to archive the REP dataset is the NetCDF (Network Common Data Form). It is self-describing since it includes the metadata that describe both data and data structures. The NetCDF implementation is based on the community-supported Climate and Forecasts (CF) specification (CF1.6 profile for profile data) and it adopts the SeaDataNet vocabularies ([https://www.seadatanet.org/Standards/Common-Vocabularies](https://www.seadatanet.org/Standards/Common-Vocabularies)). The reference SDN parameter codes (P01 terms, [https://vocab.seadatanet.org/v_bodc_vocab_v2/search.asp?lib=P01](https://vocab.seadatanet.org/v_bodc_vocab_v2/search.asp?lib=P01)) and the associated standard units (P06 terms [https://vocab.seadatanet.org/v_bodc_vocab_v2/search.asp?lib=P06)](https://vocab.seadatanet.org/v_bodc_vocab_v2/search.asp?lib=P06) are used in order to ensure the proper interpretation of values by both humans and machines and to allow data interoperability in terms of manipulation, distribution and long-term reuse.

Each XBT NetCDF file contains:

- **dimensions** that provide information on the size of the variables (a.k.a. "parameters");
- **coordinate variables** that orient the data in time and space;
- **geophysical variables** that contain the actual measurements;
- **ancillary variables** that contain the quality information (QFs) values;
- **additional variables** that include some of the variables being part of SDN extensions to CF;
- **global metadata fields** that refer to the whole file, not just to one variable (a.k.a. "global attributes").

**C.1 Dimensions**

The pattern followed by SDN for "profiles" data type is to have an 'INSTANCE' unlimited dimension plus a maximum number of z coordinate levels (*MAXZ*). We included also string size dimension STRING for text arrays and added test size dimensions referring respectively to test QFs on temperature (*TST_T*) and depth (*TST_D*) values and the maximum number of z coordinate levels for the data re-sampled at a 1 m interval, after the QC is applied (*MAX_INT*).

**C.2 Coordinate variables**

NetCDF coordinates are a special subset of variables which orient the data in time and space. They are:

- LONGITUDE for x;
- LATITUDE for y;
- TIME for t;
- DEPTH for z.

**C.3 Geophysical variables**

Each file contains:

- depth: depth at original vertical resolution;

- TEMPET01: Calibrated sea water temperature at original vertical resolution;
- DEPTH_COR: Original vertical resolution depth corrected by applying Cheng et al. (2014);
- TEMPET01_COR: Calibrated and corrected sea water temperature as resulting by applying Cheng et al. (2014);
- DEPTH_INT: depth interpolated on standard depth levels using Barker & McDougall (2020) method;
- TEMPET01_INT: TEMPET01 interpolated on standard depth levels using Barker & McDougall (2020) method;
- DEPTH_COR_INT: DEPTH_COR interpolated on standard depth levels using Barker & McDougall (2020) method;
- TEMPET01_COR_INT: TEMPET01_COR interpolated on standard depth levels (each meter depth) using Barker & McDougall (2020) method;

Calibration values are provided in a separate variable, CALIB, so that experts can trace back the raw (uncalibrated) profile if needed.

For each coordinate and geophysical variable four mandatory parameter attributes are included, as defined in Lowry et al. (2019):

1. *sdn_parameter_urn*: this is the URN (Uniform Resource Name) for the parameter description taken from the P01 vocabulary;
2. *sdn_parameter_name*: this is the plain language label (Entryterm) for the parameter taken from the P01 vocabulary at the time of the data creation;
3. *sdn_uom_urn*: this is the URN for the parameter units of measurement taken from the P06 vocabulary;
4. *sdn_uom_name*: this is the plain language label (Entryterm) for the parameter taken from the P06 vocabulary at the time of data file creation.

Moreover, since some of the coordinate variable names could be ambiguous, particularly for the z-coordinate, we adopt the standard_name (P07 vocabulary, https://vocab.seadatanet.org/v_bodc_vocab_v2/search.asp?lib=P07), not mandatory in CF but widely used, which significantly enhances interoperability.

**C.4 Ancillary variables**

In order to report data quality information on a point by point basis, every measurement is tagged with a single-byte encoded label referred to as a 'flag'. The flag variables are mandatory for all coordinate and geophysical variables to which they relate through 'ancillary_variables' in the parent variable set to the name of ancillary variable attribute (Lowry et al., 2019). The flags are encoded using the SDN L20 vocabulary (https://vocab.seadatanet.org/v_bodc_vocab_v2/search.asp?lib=L20) and each ancillary variable carries attributes 'flag_values' and 'flag_meanings', which provide a list of possible values and their meanings.

For coordinate variables, the ancillary variables are the following:

- TIME_SEADATANET_QC: it is the ancillary variable referring to TIME parent variable;

- POSITION_SEADATANET_QC: Longitude and latitude flag variables are combined into a single flag for 'position', following OceanSITES (2020) practice.

For depth coordinate, the ancillary variables are:

- DEPTH_TEST_QC: it contains flags coming from the application of depth check test;
- DEPTH_FLAGS_QC: it contains flags associated with each original depth value and summarizes the results of the performed depth test check mapped on SDN L20 vocabulary;
- DEPTH_COR_FLAGS_QC: it contains flags associated with each corrected (Cheng et al., 2014; CH14) depth value;
- DEPTH_INT_SEADATANET_QC: it contains flags associated with the interpolated profile;
- DEPTH_COR_INT_SEADATANET_QC: it contains flags associated with the corrected (CH14) interpolated profile.

For temperature geophysical variable, the ancillary variables, similarly to depth coordinate, are the following:

- TEMPET01_TEST_QC: it contains exit values coming from the application of independent temperature check tests;
- TEMPET01_FLAGS_QC: it contains the QFs associated with each calibrated temperature value and summarizes the results of the performed independent temperature test checks mapped on SDN L20 vocabulary;
- TEMPET01_COR_FLAGS_QC: it contains the QFs associated with each calibrated and corrected (CH14) temperature value;
- TEMPET01_INT_SEADATANET_QC: it contains QFs associated with the temperature interpolated profile;
- TEMPET01_COR_ INT_SEADATANET_QC: it contains QFs associated with the corrected (CH14) temperature interpolated profile

**C.5 Additional variables**

In addition to attributes, some variables from the SDN extension have been adopted:

1. *SDN_CRUISE*: an array containing the name of project which funded the cruise;
2. *SDN_EDMO_CODE*: an integer array containing keys identifying the organization in the European Directory of Marine Organizations (EDMO, https://www.seadatanet.org/Metadata/EDMO-Organisations)
3. *SDN_BOT_DEPTH*: a floating-point array holding bathymetric water depth in meters where the sample was collected or measurement was made. We considered the local bottom depth extracted from the GEBCO Compilation Group (2021).

In order to preserve and keep track of metadata associated with each profile (*ulr_metadata*) in the dissemination through ERDDAP, other variables have been adopted:

4. *cruise_id*: an array containing the name of the project which funded the cruise plus the year and the month of the cruise;

5.   *profile_id*: an array referring to the sequence of the profile during the corresponding cruise.

**C.6 Global metadata fields**

The global attribute section of a NetCDF file describes its content overall. All attributes should be human-
readable and contain meaningful information for data discovery and re-use. Most importantly, all available
discovery metadata to the SDN mandatory attributes have been introduced following recommendations of the
XBT community. Moreover, several studies (Cheng et al., 2014; 2016; 2018; Goni et al., 2019) highlighted
the dependency of the biases on probe type, time (due to variations in the manufacturing process) and changes
in the recording systems (Tan et al., 2021). For these reasons, the following information has been inserted in
the XBT metadata description: probe type with serial number, manufacturer, manufacturing date, FRE
coefficients used to calculate the depth, launch height, DAQ model and recorder version (Cheng et al., 2016).
Ship speed, wind speed, and probe mass (available since 2018) have been added to this metadata section, when
available.
The depth (depth_uncertainty) and temperature (TEMPET01_uncertainity) uncertainties, being equal to each
profile within the REP dataset, have been included as global attributes.
The above-mentioned information has been kept and made available through ERDDAP by an *url_metadata*
variable in order to manage more efficiently the many metadata strings. A Jupyter notebook in Python
(Fratianni and Frizzera , 2024) has been stored on GitHub repository and published on Zenodo
(https://doi.org/10.5281/zenodo.13862792) to access and recombine all data and metadata in NetCDF files,
one per XBT profile.

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
