# Peer review of "Reprocessing of XBT profiles from the Ligurian and Tyrrhenian seas over the"

_Earth System Science Data, 2023_

## Author Comment (AC1)

**Reviewer 1**

**General comments:**

The manuscript describes the re-processing of historical XBT data collected in the Mediterranean Sea from 1999-2019. The reprocessing involved not only creation of automated quality control procedures, addition of a test canister 'calibration' offset and new linear interpolation method, but addition of complete metadata information. The addition of accurate metadata is vital for end users to be able to correct data as new research becomes available.

The presentation of data via the ERDDAP server system makes it accessible to all, although there are some improvements to the data format that need to be made (see comments below). The text itself is well written, but could do with some grammar checks and re-wording to make quite complex sentences simpler. I have given some specific comments on this, but it doesn't cover all of them. Overall, the structure of the manuscript is very good and quite thorough.

The re-processing of such a valuable historical dataset is critical for accurate estimates of ocean heat content and improved accuracies of data assimilation into models. I recommend the publication of this paper after some changes to the text and the data files.

Answer

We thank the referee for the detailed review and the constructive comments to both the manuscript and the dataset that gave us the possibility to substantially improve the quality of our research. The specific answers to each comment are reported below and a new REP dataset version is provided at https://doi.org/10.13127/rep_xbt_1999_2019.2

**Manuscript specific comments:**

A big improvement to the dataset and manuscript would be the inclusion of uncertainty values with the data. Uncertainties are mentioned, but there is no development of or inclusion of the uncertainty data. Some brief discussion of derivation of the uncertainties should also be included.

A: We thank the reviewer for this suggestion. We decided to add the uncertainty specification based on the nominal instrument accuracy provided by the manufacturer, in agreement with Atkinson et al. (2014) and Cowley et al. (2021). The depth and temperature uncertainties are equal for all REP XBT profiles being gathered with probe types produced by Sippican, so we inserted them in the file global attributes (please check it here http://oceano.bo.ingv.it/erddap/info/REP_XBT_1999_2019_v2_metadata/index.html) to not make the dataset heavier.

| attribute | NC_GLOBAL | depth_uncertainity | String | depth<=230m: 4.6m;depth>230m: 2% (Table 2 from Cowley R et al., 2021 https://doi.org/10.3389/fmars.2021.689695) |
|---|---|---|---|---|
| attribute | NC_GLOBAL | TEMPET01_uncertainty | String | XBT = 0.10 deg C; XCTD = 0.02 deg C (Table 2 from Cowley R et al., 2021 https://doi.org/10.3389/fmars.2021.689695) |

Line 139-145 discusses measurement accuracy and uncertainties. A comment here about using manufacturer accuracies as an estimate of uncertainties would be beneficial, rather than use both terms interchangeably.

A: We thank the reviewer for highlighting this important aspect, we modified the text of Section 2.

For further corrections to the dataset, did you consider implementing the launch height correction from Bringas and Goni, 2015? You have all the launch heights from the vessels and by adding the

offset to the depths, you will get quite a different result in your comparisons. Bringas and Goni's results are robust and it will be a first for these corrections to be implemented in your dataset. Some of the vessels are 25m launch heights which equates to a 4m depth offset, a considerable impact on the thermocline depths. You could make the correction and include it in the data file to allow the user to remove it if required, in the same way as the calibration value has been done.

A: The algorithm developed by *Bringas and Goni (2015, hereafter BG15)* corrects the depth value calculated by standard FRE when the deployment height is different from about 3.0-4.0 m. BG15 approximates what happens in operational conditions. In fact, regardless of the mode and the probe impact speed with the sea surface, BG15 correction does not take into account the turbulence produced by the ship's wake in the water column crossed by the probe in the near surface layer, a perturbation that depends on the ship speed, its draft and the distance of the impact point from the side of the hull. The majority of the ships used to collect the REP datasets has operational speed greater than 20 knots (see Figure R3 below and the relative answer).

We found in the literature an unpublished communication from *Gilson, Roemmich and Johnson (2008)* that illustrates what happens when XBTs are launched from ships moving at different speeds, but it does not include any specific description about their behavior in the surface layer.

In our opinion, BG15 is a proper correction when the ship speed is close to zero, which is not the case for the majority of profiles in the REP dataset (Figure R1), so we preferred not to apply it to the REP dataset depth values and further investigate this issue in our next studies.

Figure R1 shows the distribution of launch heights per probe type, indicating that about 70% of the drops were from platforms at ~ 10-11 m height.

[Figure]

**Figure R1 - Distribution of XBT launch heights above the sea level per probe type in the REP dataset.**

You also could implement the fall rate corrections from Cheng et al 2014. Although this could be complicated if the launch height corrections are also implemented, I'm not aware of any investigations into how these two corrections would interact with each other.

A: We agree to apply the Cheng et al 2014 correction (CH14) and include it in the dataset as a separate variable. This new variable is also interpolated at each meter. In this way the user will have the possibility to consider the corrected version (CH14) of the profile according to his needs. We decided instead to not apply BG15 as motivated in our previous answer, avoiding any interaction with CH14.

Line 171-175: Does this mean that the test canister results were applied to the strip chart recording system at the time of data collection as an offset? Or were they simply used as a check, as is currently the case? I don't know if it is necessary to do this correction at all. My understanding is that the test canister allows you to check the system is earthed correctly, any failure in earthing will result in poor data collection and the test canister will show this. Also to show any faults in the launcher cable, launcher gun etc. Are there any references to suggest that the test canister offset should be applied (eg Reseghetti et al 2018, others)?

A: We would like to precize that the sentence was referring (not clearly) to the first years of SOOP activity in the Mediterranean Sea during MFS-PP and MFS-TEP projects and generally before 2008. As far as we know the test canister was used as a check. In general, this offset is not included during routine data collection but during data post-processing.

*Bordone et al. (2020)* and *Raiteri (2023*, 1. P-11 Raiteri.pdf) confirmed the improved agreement between XBT and reference profiles (CTD or Argo) evidenced by Reseghetti et al. (2018) using XBT profiles from SOOP activities. The application of the test canister correction has been suggested by field comparisons XBT vs. CTD (*Reseghetti et al., 2018*): XBT values showed a T difference nearly coincident with the test canister offset when compared with the corresponding CTD profiles.

The corresponding text in the manuscript has been modified.

Line 200: Great that you've got some numbers for evaluation of disposal of these materials at sea. Instead of including it in the Summary/Conclusions section only, it would be good to have a few sentences in the results section perhaps about the total amounts for this dataset. Or even in table 1, which I think only has values on a per probe basis. The table headings need to indicate it is per-probe, not total.

A: The sentence has been moved at the end of Section 3, where the dataset is described. The other suggestions have been also considered and included in the new manuscript version.

Line 246: What is the 'evident "noise"' referring to? Is it electrical interference (high frequency noise) perhaps? Perhaps it was an earthing fault? Did your AQC pick up this noise in a particular test? Perhaps a quick sentence suggesting a source for the noise or what it looked like? Or remove this reference.

A: The new manuscript version adds details and provides a better description of the problem. We had to deal with noise of different origins: one external connected to atmospheric factors (primarily the wind) and one intrinsic to the specific acquisition system used (Figure R2), which manifested itself with "value variations" of very different amplitude and frequency that our algorithms could not capture. Figure R2 shows the behavior of nearly collocated XBT profiles recorded with two different DAQs within a few days.

[Figure]

**Figure R2 - Comparison between profiles recorded with two different DAQs in the same area (south Tyrrhenian sea) about twelve days apart. The black and blue lines are from an MK21-ISA with problems with its electronic components while the orange and green lines are from an MK12.**

Line 307: Change 'indexes' to 'flags' and 'those repeated during data taking' to 'duplicates'. Or is it 'replicates'? If duplicates (ie, the same data copied), these should definitely be removed. If replicates (ie, two different profiles taken close together in time), these should not be removed. I also think removing data that fails in less than 50m is not the right thing to do. The data should be flagged where it is bad, but if it contains some good data, it should be kept.

A: We changed 'those repeated during data taking' to 'replicates' since the profiles have been repeated as soon as the operator realized that they had evident problems. We decided not to include these profiles because they would have been labeled with all BAD quality flags. We instead decided to integrate the dataset with those profiles having good data only within the first 50 m of depth, as suggested by the reviewer.

Line 322-326: Is there any attempt to correct positions or do you lose these data because they are 'on land'? Since you have re-processed from the start, you should be able to correct positions for at least some of the profiles. There is no mention of a date/time check (or ship speed check) - was this investigated?

A: There are not profiles are on land in the REP dataset, since the operators checked both the position and the launch time before the data transmission to the ENEA-STE. Since we did not encounter specific issues with date/time we did not implement additional checks.

Table 2 and Surface Check: The exit values of 49-52 are not very descriptive compared to the other tests. I would suggest updating them to include 'Surface check' or similar so it doesn't get confused with the SDN flag codes and is clearer for the user of the data. It will mean updating the data file variable attributes too.

A: We thank the reviewer for the suggestion but we prefer to keep these exit values in the data files modifying their meaning in Table 2.
49: T difference < 1 SD
50: 1 SD < T difference < 2 SD
51: 2 SD < T difference < 3 SD
52: T difference > 3 SD

Line 515-516 and Figure 7. The intervals described in the text are not clear in the figures. The bias figures perhaps need some lines showing these intervals? And I think that the figure needs 'Dec-May' and 'Jun-Nov' instead of 'mixed' and 'stratified' to be consistent with the text. Line 518 should also refer to the month period rather than 'mixed'. The brackets in the x-axis label are not clear.

A: The figure and the text have been modified as suggested by the reviewer.

Line 647-649: the depth in figure 12c is 500 to 800 m and shows a negative average bias, not consistent with the positive bias in the deeper part of figure 11a (which is 800-2000m and solely due to T5 probes). I don't think the steps have any impact on the average bias, but do cause the positive spikes in the figure 12c T difference plot.

A: We thank the reviewer for pointing this out. We further checked the deep profiles between 800-2000m and we agree that the steps do not have an impact on the average bias, which is mainly due to the calibration. The text and figure 12 have been modified accordingly:

"*Figure 12 shows examples of matching REP and SDN profiles and their difference with a zoom in the surface (a) and bottom layer (b and c), where the largest differences occur. During the stratified period, the largest differences reside in the thermocline and can exceed 1.5 °C (Figure 12a), while in the bottom layer the calibration correction (see Figure 12b, c) together with the abrupt decrease of the number of data explain the small positive average bias in Figure 11a. In fact, numerous T5/20 profiles were launched (~7% of the total) in the few campaigns in which the acquisition system showed significant negative anomalies (up to - 0.1 °C, see Figure 5a) and this influenced both BIAS and RMSD profiles below 900 m depth. The frequent step-like shape of deep profiles (Figure 12c), due to double diffusion processes (Meccia et al. 2016; Durante at al., 2021), causes instead positive spikes in the difference profiles.*"

**Editing corrections/grammar suggestions:**

line 24: insert SDN acronym after the Seadatanet name.

A: Done

Line 25: The link does not work when clicked, seems to be a different link to what is written.

A: We apologize for this. We checked and clicking from the web page the URL works https://essd.copernicus.org/preprints/essd-2023-525/#discussion, it does not work from the downloaded pdf. We copy it here https://cdi.seadatanet.org/search/welcome.php?query=1866&query_code={4E510DE6-CB22-47D5-B221-7275100CAB7F}

Line 29: June to November perhaps, as written later in the text?

A: We thank the reviewer for noticing this, we corrected it.

Line 34: Is there a full name for the ERRDAP acronym? If so, include here.

A: Done

Line 44: 'to re-analyze' change to 'reanalysis of'

A: Done

Line 68: WOD link does not work, doesn't match the text.

A: We apologize for it but we do not understand why this happened, we will double check all the links in the revised manuscript before submission.

Line 66-70: this sentence is missing something grammatically, please review.

A: The sentence has been rephrased: "This data review originated from the recognition that the historical XBTs from the Ligurian and Tyrrhenian Seas, presently available in the main marine data infrastructures - SDN (https://www.seadatanet.org/), WOD (https://www.ncei.noaa.gov/products/world-ocean database), Copernicus Marine Service (CMS, https://marine.copernicus.eu/) - have incomplete metadata description and the data might also differ."

Line 85: Add full words for DAQ acronym (Data Acquisition System)

A: Done

Line 88: is there a reference for latest documented QC procedures?

A: We have added Cowley et al. (2022), Parks et al. (2022), Good et al. (2023) and Tan et al. (2023) references.

Line 99: link fails when clicked.

A: We apologize for it but the URL is correct and we do not understand why this happened. We will double check all the links in the revised manuscript before submission. https://progetti.ingv.it/it/progetti-dipartimentali/ambiente/macmap

Line 119: Add "Negative Temperature Coefficient" for NTC acronym

A: Done

Line 124: '1960's' is stated as 1990's in appendix A, Line 776. Please check, probably it's since probes were built, so 1960's is likely correct.

A: We thank the reviewer for noticing it. The text in Section 2 has been modified to explain and justify the indicated periods.

Line 126: change 'clockwise in' to 'clockwise from' and 'counterclockwise in' to 'counterclockwise from'

A: Done

Line 127: suggest rewording to 'decouples the XBT vertical motion from the translational …

A: Done

Line 131: suggest removing 'phenomenological', not required here.

A: Done

Line 135: replace 'thus' with 'then'

A: The phrase has been slightly modified including your suggestion:

"The software transforms a time series of resistance values sensed by the thermistor into a series of depth - T values using first a resistance-to-temperature conversion relationship (identical for all XBT types because it is specific for the thermistor used, see Appendix A) and then calculating the corresponding depth values by applying a specific FRE for each probe type."

Line 144: 'and slightly' doesn't make sense in this sentence.

A: The text has been modified.

Line 146-150: This sentence is too long and confusing and I'm not sure what the purpose is of including it is. I also don't understand 'in order to have a practically unchanged measurand' in this sentence. Please review.

A: The text has been modified as suggested by both reviewers and new details have been inserted to answer some of their questions:

"Bordone et al. (2020) compared XBT profiles from SOOP activities in the Ligurian and Tyrrhenian Sea with quasi contemporaneous (± 1 day) and co-located (distance smaller than 12 km) Argo profiles. The XBT profiles used by Bordone et al. (2020) are included in the REP dataset but they went through a different QC and interpolation procedure that could slightly modify their results. In the 0-100 m layer, the mean T difference was 0.24 °C (the median 0.09 °C) and the Standard Deviation (SD) was 0.67 °C. Below 100 m depth, the XBT measurements were on average 0.05 °C warmer than the corresponding Argo values (mean and median were almost coincident) and the SD was 0.10°C. This last SD value agrees with the manufacturer specification and the T uncertainty value reported by Cowley et al. (2021), which has been assigned to the REP data. The values estimated by Bordone et al. (2020) for the surface and sub-surface layer (depth < 100 m) are instead affected by both the XBT (4.6 m) and Argo (2.4 dbar) depth uncertainty estimation, meaning that a small variation in depth could correspond to a large variation in temperature especially when the seasonal thermocline develops, so that the comparison with Argo values would not be significant. The specified uncertainties are independent of the systematic error or bias affecting the XBT temperature and depth measurements, that have been corrected in the REP dataset applying the Cheng et al. (2014) correction scheme."

Line 150: Include 'standard deviation' with SD acronym

A: Done

Line 152: I disagree that it is a 'few tens of meters', equivalent to more than 20 meters. The reference states only a few meters to reach stable velocity.

A: Thanks for the correction, it is our mistake in defining the range within which the phenomenon occurs. The text has been modified.

Line 159: replace 'depends on specific FRE with actual' to 'has'

A: Done

Line 160: remove 'reading'

A: Done

Line 164-166: This sentence is a little confusing, please review.

A: The sentence has been rephrased. "The computer clock, always updated to the UTC value before and after the data gathering, provides the time coordinate of each profile with a sensitivity of 1 s. The differences recorded with respect to the UTC standard time have always been no greater than 1 s over a 24 hour time frame."

Line 170: 'is binding for subsequent optimal use' could be better written

A: The sentence has been rephrased: "... is highly recommended for an optimal use of XBT measurements."

Line 171-175: the grammar is quite awkward and could be better written.

A: The sentence has been rephrased: "When strip chart recorders were used, a preliminary and accurate calibration of the acquisition unit using a tester was mandatory (e.g. Sippican, 1968 and 1980; Plessey-Sippican, 1975). With the advent of digital systems this procedure was also recommended (Bailey et al., 1994)."

Line 182: Are you using accuracy and uncertainty terms interchangeably here?

A: We apologize for the improper use of the terms, we modified the text to avoid confusion.

Line 184: suggest changing 'thanks to a' to 'with a'

A: Done

Line 185: remove 'of'

A: Done

Line 189: remove 'of'

A: Done

Line 193: Why would ship speed affect the duration of the data acquisition? The probes are designed to take the ship speed out, as you have stated earlier.

A: We report hereafter an additional analysis of the ship speed versus the depth of the last good T value to show the influence of the ship speed on the duration of the acquisition (equivalent to the length of the profile).

Figure R3 shows the distribution of the ship speed at the XBT launch time per probe type for the REP dataset, which reflects the characteristics of the vessels used. The GNV ships, from which most of the probes were launched, usually travel between 21.5 and 23.5 knots, a speed higher than the nominal values for all XBT types, except the T4 model. The ship speed can influence not only the duration of the acquisition (i.e. the maximum depth achievable with acquisition software in full acquisition mode) but also the quality of the recording, which degrades at very high speed.

Figure R4 shows two DB profiles gathered at a distance of ~ 12 nm but at different ship speeds: the profile collected at higher speed is shorter (724 m instead of 850 m) and noisier than the other one.

Figure R5 shows the mean and the median of the depths associated with the last good T value (Last Good Depth) for DB probes as a function of the ship speed. It is evident the shortening of the profile when the ship speed increases. The anomaly at 17-18 knots can be explained by cruises with

container ships having a very high launching platform (25 m or more) associated with bad weather conditions

[Figure]

**Figure R3 (Left panel) Distribution of the XBT probes used in the REP the dataset as a function of the ship speed. (Right panel) zoom.**

[Figure]

**Figure R4 - Two DB profiles gathered consecutively but at different ship speeds.**

[Figure]

**Figure R5 The mean and the median of the depth associated with the last good temperature value for DB probes (dropped with software set as free terminal depth) as a function of the ship speed.**

Line 198: Include full name for ZAMAK

A: Done

Line 266: remove 'only'

A: Done

Line 280: change 'masses' to 'mass'

A: Done

Line 290: add 'Delayed Mode' for DM acronym

A: It is already specified in the Introduction.

Line 289 & 292: Link fails when clicked

A: We apologize for it but the URLs are correct and we do not understand why this happened. We will double check all the links in the revised manuscript before submission.

Line 306: change 'eventually remove it' to 'remove it if required'. Also change 'profiles' to 'data' and 'eliminated' to 'deleted'.

A: Done

Line 310: remove 'implemented'

A: Done

Line 316: Suggest changing to 'Automated Quality Control overview' or 'Automatic Quality Control procedure'

A: Done

Line 319: Change 'flag' to 'exit value' to be consistent with Table 2.

A: Done

Line 325-326, 329: suggest using the same terminology as is in table 2, rather than 'GOOD' and 'BAD'.

A: Done

Line 355: Change 'It' to 'The Gross range check'

A: Done

Line 343-345: suggest changing to "The XBT measurements close to the sea surface are usually considered unreliable due to the time taken to reach terminal velocity (Bringas and Goni, 2015) and due to the time taken for the probe to reach thermal equilibrium (need a reference here) and are thus excluded from further analysis (e.g. Bailey et al., 1994; Cowley and Krummel, 2022).

A: The new sentence has a modified structure (with additional details) in order to better describe the problem and the proposed way out.

Line 345-347: Suggest changing to: 'We here implement a surface test that flags data and retains all original measurements.'

A: This suggestion has been included in a new sentence.

Line 347: remove 'proposed' as you have implemented it.

A: Done

Line 348: need a reference for the 'first value currently considered acceptable'

A: New version includes a historical note and references.

Line 390: should be figure 4a

A: Done

Line 467: suggest removing 'k-th' as it's confusing. Or re-write?

A: We have substituted 'k-th' with 'k' in the 4.3.1 section

Line 489-490: suggest changing 'crossing the water column and measuring' to 'deployed' or 'recording'

A: New sentence has been prepared.

Line 491: suggest removing '("hot" or "cold" probe or possible troubles during the acquisition)' as it is unnecessary and leads to questions about what is meant by hot and cold probes.

A: The new version avoids these terms.

Line 545: Suggest 'The QC algorithms applied to the dataset are not capable of catching all erroneous values.'

A: Done

Line 549: remove 'deeply'. Change 'by visual check' to 'using visual checks'. Change 'In specific' to 'Specifically'.

A: Done

Line 550: Change 'tuned by' to 'using'

A: Done

Line 551: change 'minimize to flag as BAD data the GOOD ones.' to 'minimize flagging of BAD data as GOOD.'

A: Done

Line 553: 'from a visual'

A: Done

Line 555: change to 'flagging of GOOD data as BAD, as shown....'

A: Done

Line 558: remove 'instead'

A: Done

Line 559: change to 'true positive spikes (a) and false positive spikes (b)'

A: Done

Line 561: 'features that the automatic'

A: Done

Line 563: remove 'happened or'. Change 'The indispensable premise is the' to 'The decision is based on the'

A: Done

Line 575: suggest removing 'non-zero' as if there is wind it is of course non-zero

A: Done

Line 578: remove 'also', already used earlier in the sentence.

A: Done

Line 582: remove '"cleanliness" of the'

A: Done

Figure 8 & 9 titles have underscores in them which have turned text into subscripts. They are also a bit cryptic for the reader, perhaps useful in analysis but could be improved for the manuscript.

A: Done

Figure 8 caption for (b) should 'true' be 'false', as in the text?

A: Thank you for noticing this, we have corrected it.

Figure 9 caption: change '(a) true spikes; (b) false spike' to '(a) true positive spikes; (b) false positive spike'

A: Done

Line 599-608: too many words in quotations. Suggest removing all the quotation marks.

A: Done

Line 599: change to '...identify the external influences that cause high frequency noise in the T profile...'

A: Done

Line 606 - 607: change to: 'In some cases, the automated QC BAD attribution was changed to GOOD after the comparison with adjacent profiles that present similar characteristics.'

A: Done

Line 612: hyperlink is not the same as the text

A: We apologize for this inconvenience, we will double check them in the revised manuscript.

Figure 10: I suggest these axes be re-shaped to portrait mode and increase the scale for temperature to avoid viewing the 0.01 resolution steps in the data. The way they are set out at the moment makes it very difficult for the reader to see the features that are talked about. The main focus is the noise, not the inversions.

A: A revised figure has been inserted.

Line 631: 'profiles without correction' and 'non-corrected profiles' are the same thing. One should be 'corrected'.

A: We thank the reviewer for noticing this, we have corrected: "The bias is larger (~0.06 ºC) when estimated from profiles without calibration correction and slightly smaller (~0.04 ºC) from calibrated profiles..."

Line 635: change 'quite constant' to 'consistent'

A: Done

Figure 11: What is the black line in the plots? Only two of the colours are referenced, but there are three.

A: Thank you for pointing this out, we added the legend to subplot (a).

Line 645: is 'relative' differences referring to the dt/dz within a profile?

A: The phrase has been corrected "Figure 12 shows an example of matching REP and SDN profile and their difference". The caption of Figure 12 has also been modified "...(a) whole profiles on the left and their difference on the right;..."

Line 669: 'The adoption of a Gaussian filter...' is this referring to the SDN dataset?

A: Yes, it is. We specified it: "The adoption of a Gaussian filter in SDN data (Manzella et al., 2003; 2007) ..."

Line 736: Rebecca Cowley is not a Dr.

A: Done

Line 758-759: remove 'because it is an essential component to get good quality XBT measurements.'

A: Done

Line 822: is 'URN' defined anywhere, if not please define.

A: Thanks for noticing it, we added the definition URN (Uniform Resource Name).

Line 870 & 879: Remove 'Moreover'

A: Done

Line 884-886: change to: 'Ship speed, wind speed, and probe mass (available since 2018) have been added to this metadata section, when available.' Remove the rest.

A: Done

**Data file comments:**

Thank you for including raw data and the temperature calibration values that can be subtracted. This makes the data file versatile for the user.

- I suggest replacing the 'TEMPE01' name in the variables with 'TEMPERATURE' to make it easier to read.

A: We understand the reviewer's suggestion but the variable name comes from the use of P01 SDN vocabulary, we cannot modify it. Please look at https://vocab.seadatanet.org/v_bodc_vocab_v2/browse.asp?order=conceptid&formname=search& screen=0&lib=p01&v0_0=TEMPET01&v1_0=conceptid,preflabel,altlabel,definition,modified&v2_0= 0&v0_4=&v1_4=modified&v2_4=9&v0_5=&v1_5=modified&v2_5=10&x=0&y=0&v1_6=&v2_6=&v1 _7=&v2_7=

Is the TEMPET01_TEST_QC variable additive? I suspect so since there are values of 581 in the variable, which means that more than one test is failed at a given depth. If so, please review this

information about when to use 'flag_values' and 'flag_masks' attributes. I think it needs to be 'flag_masks' if it is additive. And, the 'flag_values' attribute should be bit values that can be decoded unambiguously into the individual 'flag_meanings' associated with each bit.

A: We thank the reviewer for pointing this out. The TEMPET01_TEST_QC variable includes the exit values of all QC tests, with each column corresponding to a test output, as detailed in Table 2. Flag values of 581, 582 or 571, 572 are assigned from the vertical gradient check, which is applied 3 times in an iterative way. Measurements with corresponding out of range vertical gradient values are flagged (58 or 57 depending on the gradient sign) and discarded when vertical gradient values are re-computed. 581 and 571 flags are used during the second loop check, 582 and 572 during the third loop check. TEMPET01_TEST_QC is then used to map the tests' exit values to the ancillary variable TEMPET01_FLAGS_QC, as explained in section 4.2. Moreover, for the TEMPET01_TEST_QC and DEPTH_TEST_QC variables, we decided to follow the SDN approach and we maintained only some of the mandatory attributes (flag_values, flag_meanings) adapted to the QC tests needs.

The text in section 4.1 has been modified: "Results of each test are recorded by inserting the relative exit value to the corresponding measurement in TEMPET01_TEST_QC ancillary variable according to the scheme shown in Table 2"

The text in section 4.2 has been modified: "Each basic QC test assigns a corresponding exit value (Table 3) to each original depth (DEPTH_TEST_QC) and T record (TEMPET01_TEST_QC) within the vertical profile …"

- There are many global attributes that need to be made into variables. I downloaded a netcdf file via the ERDDAP server and it created one netcdf file with many profiles dimensioned by 'row'. That means that all of the global variables that would apply to ONE profile now do not apply and need to be made into variables. For example: fall_rate_equation_Coeff_1, fall_rate_equation_Coeff_2, probe_type information, launch_height information, serial numbers, platform codes and so on. Similarly, if I look at an ascii line dump of the data, all of the attribute names contained in 'global attributes' section of the netcdf files are missing. Please review these global attributes carefully. The text in the appendix will also need updating if these items are moved to variables.

A: An url_metadata variable has been inserted. Now the metadata associated with each profile can be retrieved through the profile_id and cruise_id variables. A python script has also been added in the Appendix C to facilitate the user.

- Attributes for the variables: please check these. Some have incorrect attributes (eg, DEPTH_*_QC variables have a 'standard_name' attribute of 'depth' where it should be '*_status_flag'). The 'TEMPE01' variables are missing a standard_name attribute ('sea_water_temperature').

A: The standard name for DEPTH_*_QC is created by ERDDAP automatically: It has been corrected and for all the variables it was inserted "status_flag". The same approach has been followed for TEMPET01_*_QC.

---

## Author Comment (AC2)

**GENERAL COMMENTS:**

The manuscript presents a reprocessed version of previously published XBT data. The previous version(s) did not have the best available calibrations and/or quality control applied, and this new version apparently does. As such, I find the effort worthwhile - it is always good to have a version of a dataset that can be considered "final".

The manuscript could be made stronger by:

1. Adding an uncertainty estimate against an independent data source (Argo?) that validates the new data to be in better agreement with such reference than the previous version;
2. Adding a use case that shows what can be done with the new version that could not already be done with the old one(s). E.g. can we detect temperature trends now with better confidence (or after fewer years) than before?

These two items should make the points that yes, the new data is better, and yes, it was actually worth the effort. The present manuscript describes the methods in sufficient detail, but does not make these points.

The clarity of the manuscript could be improved by a copy/line editor authorized to make more than just minor language editing. Can the journal provide such services, for a fee if need be?

There are substantial problems with the dataset and the metadata that comes with it. None of these problems are unusual or difficult to correct, but they do need correction. For a manuscript that lays claim to high-quality metadata, the present state of the underlying dataset is not acceptable. Comments below list my findings in detail.

*A: We thank the reviewer for the comments and suggestions, especially on the dataset which allowed us to substantially improve it. We just started our data publishing service with ERDDAP and we are still learning. Please see below our detailed manuscript and dataset review reaction, and please also have a look at our answers to reviewer #1. A new REP dataset version is provided at https://doi.org/10.13127/rep_xbt_1999_2019.2*

*We would like to precise that our objective was to release for the first time the complete dataset with comprehensive documentation of the new processing procedure. The REP dataset provides for the first time the raw profiles with calibration correction and the full metadata information (i.e. probe type, ship speed, launch height). A new automatic Quality Control and a new interpolation procedure have also been applied.*

*The XBT dataset available from SeaDataNet infrastructure consists of only interpolated profiles without calibration correction applied (Line 22) that have been quality controlled following (Line 26) Manzella et al. (2003, 2007). We performed a REP-SDN comparison to prove to the users how a different data processing (calibration, QC, interpolation) might affect the final interpolated profiles.*

*The lack of metadata information about the XBT probe type characterizes the main marine data infrastructures since in the past these metadata were not considered crucial for data re-use and integration with other data types. Cowley et al. (2021) report that only 50% of the World Ocean Database contains XBT probe type and manufacturer information, owing to the application of intelligent metadata algorithms to recreate them. The need for such information to reduce the uncertainty in the computation of the Ocean Heat Content indicator has also been widely reported in literature and it has been one of the motivations for this data review. We will revise the Med OHC estimation once the data description paper has been finalized. In fact, the present data description paper is already very long and rich in details that we would like to consolidate before any further data analysis.*

*We decided to add the uncertainty specification based on the nominal instrument accuracy provided by the manufacturer, in agreement with Atkinson et al. (2014) and Cowley et al. (2021). The depth and temperature uncertainties are equal for all REP XBT profiles being gathered with probe types produced by Sippican, so we inserted them in the file global attributes (please check it here http://oceano.bo.ingv.it/erddap/info/REP_XBT_1999_2019_v2_metadata/index.html) to not make the dataset heavier.*

| | | | | |
|---|---|---|---|---|
| attribute | NC_GLOBAL | depth_uncertainity | String | depth<=230m: 4.6m;depth>230m: 2% (Table 2 from Cowley R et al., 2021 https://doi.org/10.3389/fmars.2021.689695) |
| attribute | NC_GLOBAL | TEMPET01_uncertainty | String | XBT = 0.10 deg C; XCTD = 0.02 deg C (Table 2 from Cowley R et al., 2021 https://doi.org/10.3389/fmars.2021.689695) |

*Reseghetti et al. (2018) and Bordone et al. (2020) performed XBT-CTD and XBT-Argo intercomparisons. The XBT profiles used are included in the REP dataset but they passed through a different QC and interpolation procedure that could slightly modify the results. We consider their results on XBT uncertainties estimation valid and we plan to update them soon but without expecting substantial changes. These values are consistent with the uncertainty values that we specified in the new manuscript version (Cowley et al., 2021; Tables 1 and 2).*

*We checked with the editorial office and every paper, once accepted, receives English language copy editing. However, we tried to improve the paper readability also thanks to the reviewers' suggestions.*

SPECIFIC COMMENTS:

**Manuscript:**

Ll. 25-34: I recommend taking the URLs out of the text and putting them in footnotes. The one that breaks on the end-of-line cannot be used "as is", but requires hand-editing - that too should be corrected.

*A: We realize that the pdf conversion of the manuscript did not preserve the correct URL links. We will take care of this with the editorial office during the next reviewing steps. We will also check your suggestion of inserting URLs as footnotes with the editorial office.*

I had to read these sentences twice to understand which dataset was the original, and which was the new one that was being described in this article, and why there were three links instead of two. I recommend clarifying by e.g.:

assigning names "ORIGINAL" and "REPROCESSED" to these, and using these names throughout the manuscript
removing one of the two links to the reprocessed data (keep the doi one)

*A: We apologize if the abstract is not straightforward in describing the two dataset versions:*

- *the original dataset (raw data with calibration correction) has never been published before;*
- *the SeaDataNet dataset version (SDN) does not include raw data but only post-processed interpolated ones according to Manzella et al. (2003, 2007). It does not have calibration information and complete metadata description, i.e. no probe type, fall rate coefficients, manufacturer, ship speed, launch height);*
- *We decided to remove the SDN URL from the abstract, as suggested by the reviewer, and leave it in section 5.1;*
- *the reprocessed (REP) dataset has been prepared starting from the raw data and all available information in the operational log sheets.*

*We prefer to keep the names REP and SDN datasets and improve their explanation in the abstract.*

*We removed the third link at our ERDDAP webpage since it is confusing.*

L. 30: Bias and RMS difference against what - between the old and the new versions? Is there any evidence that the new dataset is better than the old one, i.e. that bias and RMS against the truth is now smaller?

*A: We computed the bias and RMSD between the SDN and the REP versions with the objective not to prove that one version is better than the other but that they are different due to the new Quality Control procedure, the calibration correction and the new interpolation applied. The new interpolation technique (Barker and McDougall, 2020) has been selected because it recreates values closer to the true measured ones rather than the other two methods considered (Section 4.4). The final result is that the REP profiles are different from the SDN ones, especially in the surface layer from June to November when the thermocline settles. We believe this is crucial information to the data users.*

*If the reviewer intends as the truth, the nearest Argo or CTD profiles, we did not provide this comparison here. This analysis will be included in a next paper. The aim of the present data description paper is to publish for the first time the original dataset with full metadata description, which allows the users to utilize the XBT profiles for their applications and also to test alternative Quality Control procedures. We also provide a new calibrated/ QCed and interpolated data version with complete documentation of each processing step (each QC test applied to each measurement corresponds to an exit value and all test results are then mapped to a quality flag).*

*Reseghetti et al. (2018) e Bordone et al. (2020), as data providers, used some of the original data that are also contained in the REP dataset, for their comparison with CTDs and Argo profiles, but only here these profiles are shared openly with full metadata information, allowing the actual transparency and replicability of their work. We consider their data intercomparison still valid even if the XBT data have been processed differently. We will update next their results using our REP data version without expecting substantial changes but having the awareness that our QC and interpolation procedure are completely documented.*

L. 5: Define acronym ENEA.

*A: Done*

L. 28: Define acronym SDN.

*A: Done*

L. 85: Define acronym DAQ.

*A: Done*

L. 198: Define acronym ZAMAK.

*A: Done*

L. 207: Define acronym CSIRO properly.

*A: Done*

L. 211: Define acronym CNR-ISMAR

*A: Done*

There are inconsistencies between the numbers of profiles reported in table 1 versus what is in the actual dataset. Please correct or clarify:

- Summing up the second-to-last column of table 1, I expect 3917 profiles in the SeaDataNet repository.
- Clicking on the link in the abstract led me to a data download that ultimately gave me 3662 individual files. Is there a reason these numbers do not match?
- Summing up the last column of table 1, I expect 3757 profiles. The downloaded REP dataset seems to contain 3754.

*A: We apologize for the confusion with the numbers that we clarify hereafter:*

- *Table 1 does not refer to SDN but to the REP dataset*
- *if you go to the SeaDataNet portal, using the saved query at the URL https://cdi.seadatanet.org/search/welcome.php?query=1866&query_code=%7B4 E510DE6-CB22-47D5-B221-7275100CAB7F%7D, you get 3661 profiles due to the bounding box selection which cannot filter out precisely the REP dataset*

*displayed in Figure 1. Moreover, not all the profiles in the REP dataset have been disseminated through SeaDataNet. We specify in Section 5.1 that the REP vs SDN comparison has been performed on the 3104 matching profiles.*

- *We eliminated the second-to-last column of table 1 since it is not used in the manuscript and it is confusing.*
- *The numbers in the last column of table 1 have been modified. We found a bug on cruise_id so three profiles coming from the same cruise were skipped. This issue has been resolved now. New numbers include some profiles that have been re-introduced in the dataset following the reviewer's #1 suggestion.*

Ll. 146-150: Edit this sentence/paragraph for better English language:

*A: The text has been modified as suggested by the reviewer and new details have been inserted to answer some of the questions posed earlier:*

*"Bordone et al. (2020) compared XBT profiles from SOOP activities in the Ligurian and Tyrrhenian Sea with quasi contemporaneous (± 1 day) and co-located (distance smaller than 12 km) Argo profiles. The XBT profiles used by Bordone et al. (2020) are included in the REP dataset but they went through a different QC and interpolation procedure that could slightly modify their results. In the 0-100 m layer, the mean T difference was 0.24 °C (the median 0.09 °C) and the Standard Deviation (SD) was 0.67 °C. Below 100 m depth, the XBT measurements were on average 0.05 °C warmer than the corresponding Argo values (mean and median were almost coincident) and the SD was 0.10°C. This last SD value agrees with the manufacturer specification and the T uncertainty value reported by Cowley et al. (2021), which has been assigned to the REP data. The values estimated by Bordone et al. (2020) for the surface and sub-surface layer (depth < 100 m) are instead affected by both the XBT (4.6 m) and Argo (2.4 dbar) depth uncertainty estimation, meaning that a small variation in depth could correspond to a large variation in temperature especially when the seasonal thermocline develops, so that the comparison with Argo values would not be significant. The specified uncertainties are independent of the systematic error or bias affecting the XBT temperature and depth measurements, that have been corrected in the REP dataset applying the Cheng et al. (2014) correction scheme."*

Ll. 140-145: These quoted uncertainties are consistent with each other. State so.

*A: We thank the reviewer for this suggestion, the text has been modified accordingly.*

Ll. 146-150: Are these uncertainty estimates for XBT data using the dataset presented here, or are these different data? Has such a comparison been made for the data presented here?

*A: The text has been modified and new details have been inserted to answer this question. Please consider also our previous answers.*

L. 311: What does the word "imported" mean? Imported where?
L. 312: What does the word "collection" mean? If there is a specific meaning that only ODV users can understand, please explain.

*A: We thank the reviewer for pointing this out, both words "imported" and "collection" are specific terms of ODV functionalities. We deleted the phrase in the revised manuscript to avoid confusion.*

**Dataset:**

The dataset comprises ~3800 ocean temperature profiles (i.e. observations of temperature as a function of depth). This is not a particularly large dataset, and it is therefore reasonable that a user would like to download everything at once. From a quick back-of-the-envelope calculation, I assume that the size of the entire dataset (incl. metadata) should be a few hundred megabytes. However, when I tried to download the entire dataset with the settings below, the server failed with either error message 500 or 502. I assume it ran out of memory when I requested:

> http://oceano.bo.ingv.it/erddap/tabledap/REP_XBT_1999_2019.html
> requesting every variable
> requesting full time period (1999-2019)
> requesting either file type .ncCF or .ncCFMA

*A: We thank the reviewer for this important comment. We increased the RAM of the dedicated Virtual Machine and improved its set up.*

I then downloaded subsets (final ~6 months) of data, and these files do not make prudent use of memory (100-300 MB for 34 profiles). In particular, data types were unnecessarily large (e.g. floating-point variables when smaller integers would suffice), and there were many cases where information was redundant (e.g. ship names repeated for every data point, rather than once per profile). I recommend making changes that will reduce the total file size to less than ~500 MB, such that a user can "get everything at once". I recommend the following changes to save space (but please use your own good judgment - not all of this might work as intended):

Convert to 8-bit integers (instead of 16): DEPTH_FLAGS_QC, POSITION_SEADATANET_QC, TEMPET01_FLAGS_QC, TIME_SEADATANET_QC. You only ever have values 0-9, why make space for 32000?

*A: We made the changes suggested by the reviewer.*

Convert to 32-bit floats (instead of 64): depth, DEPTH_INT, TEMPET01, TEMPET01_INT. Single-precision is sufficient for temperature (~millionth of a degree) and depth (~0.1 mm).

*A: Done*

Convert to 16-bit integer (and apply corrections below) to TEMPET01_TEST_QC

*A: Done*

The following variables should have the same dimensionality as latitude and longitude (i.e. one per profile; should not be repeated for every data point):

POSITION_SEADATANET_QC, SDN_BOT_DEPTH, SDN_CRUISE, SDN_EDMO_CODE, TIME_SEADATANET_QC, cruise_id, institution,

institution_edmo_code, pi_name, platform_code, platform_name, platform_type, source, wmo_platform_code, url_metadata

*A:Done*

Consider eliminating the following: DEPTH_INT_SEADATANET_QC and TEMPET01_INT_SEADATANET_QC (the ...INT variable doesn't really need a QC flag, assuming that only "good" input data were used for the interpolation) DEPTH_TEST_QC (you already have DEPTH_FLAGS_QC, one is enough)

*A: A fundamental requirement for interoperability is the understanding of data quality on a point by point basis. This is achieved by tagging every measurement with a single-byte encoded label (flag) incorporated as CF ancillary variables. These are linked to the geophysical variable through the "ancillary_variables" attribute in the parent variable set to the name of the ancillary variable.*

*In TEMPET01_INT_SEADATANET_QC we used a flag for "interpolated_value" to highlight when the profile has been "reconstructed" in correspondence of layers larger than 3m with BAD or PROBABLY BAD (not used in the interpolation) data in the calibrated profile.*

*DEPTH_TEST_QC includes all the QC test exit values, while in DEPTH_FLAGS_QC the test exit values have been mapped to quality flags. This is done analogously for TEMP01_TEST_QC and TEMP01_FLAGS_QC.*

area should be a single global attribute, not a 17*Nobs array (!!!)

*A: Done*

The profile_id variable should be replaced with a 16-bit integer in ragged array representation (instead of 136 bits of redundant text)

*A: Done*

The naming of the variables in the dataset can be improved: some names are capitalized, others are not. Can you make all the same, or have a logic which ones are capitalized (e.g. the lat/lon/time coordinates plus depth and temperature)?

*A: ERDDAP manages the spatial and temporal features of each dataset in such a way that they have specific names and units. This makes it easier to identify datasets with relevant data, to request spatial and temporal subsets, to make images with maps or time-series, and to save data in geo-referenced file types (e.g., .esriAscii and .kml). In tabledap, a depth variable (if present) always has the name "depth" and the units "m" below sea level. Locations below sea level have positive depth values.*

TEMPET01 seems to be the primary scientific variable, but its name is not human-readable. Can this be changed to "TEMPERATURE"?

*A: We understand the reviewer's concern but the variable name comes from the adoption of P01 SDN vocabulary, we cannot modify it. Please have a look at https://vocab.seadatanet.org/v_bodc_vocab_v2/browse.asp?order=conceptid&formname=search&screen=0&lib=p01&v0_0=TEMPET01&v1_0=conceptid%2Cpreflabel%2Caltlabel%2Cdefinition%2Cmodified&v2_0=0&v0_4=&v1_4=modified&v2_4=9&v0_5=&v1_5=modified&v2_5=10&x=0&y=0&v1_6=&v2_6=&v1_7=&v2_7=*

Some variables seem to copy input data from SeaDataNet. If they are just duplicates (I have not checked if they are), is it really necessary to include these here? If we want to include them, can they at least be named consistently (at present, some start with "SDN_..." while others have "...SEADATANET..." somewhere in the middle)?

*A: None data from SeaDataNet are copied in the REP dataset. We adopted SeaDataNet standards and vocabularies that suggest these variables' names.*

There is some poor wording in the metadata of the primary temperature data, which ought to be improved. This is about the use of the words, "raw" and "calibrated". In my understanding, TEMPET01 is calibrated data at the original resolution in space/time, and TEMPET01_INT is data with the same calibration but interpolated onto a consistent depth grid. There are no "raw" data in these files. How about:

TEMPET01: long_name = 'Calibrated seawater temperature at original vertical resolution'

TEMPET01_INT: long_name =' Calibrated seawater temperature interpolated on standard depth levels'

The 'comment' attributes under CALIB and TEMPET01 should reflect this wording (i.e. get rid of "raw"). Also, the equations shown in the 'comment' attributes should use the actual variable names.

*A: The metadata have been corrected as suggested by the reviewer, please have a look here* [http://oceano.bo.ingv.it/erddap/info/REP_XBT_1999_2019_v2_metadata/index.html](http://oceano.bo.ingv.it/erddap/info/REP_XBT_1999_2019_v2_metadata/index.html)

In the CF conventions, the attribute "standard_name" is the preferred mechanism by which a user (human or computer) finds out which physical quantity is in a variable. Therefore, I recommend that all variables for which such name definitions exist, should use one. In the present version, this is not Done consistently. In particular:

- All temperature variables should have a standard_name attribute set to, "sea_water_temperature"
- The variables DEPTH_FLAGS_QC, DEPTH_INT_SEADATANET_QC, DEPTH_TEST_QC have a wrong standard_name. Should be corrected as per next item.
- All QC variables should have one of two options for standard name: either simply, "quality_flag" or the standard_name of the corresponding data variable, followed by " status_flag", as in, "sea_water_temperature status_flag" or, "depth status_flag"

*A: The standard_name is not mandatory in SeaDataNet guidelines but we added it to each QC variable as suggested by the reviewer.*

I was expecting the depth and DEPTH_INT variables to have different lengths (and likewise for the matching temperature data and QC flags). You could save some disk space by not zero-padding the (shorter) interpolated ones.

*A: The use of an equal TST size for all variables that require it depends on an ERDDAP limitation. ERDDAP reads the various dimensions and associates all the variables that have*

*only either INSTANCE or TST_D or MAXZ. This explains why DEPTH_INT and "depth" have the same lengths.*

TECHNICAL CORRECTIONS:

**Manuscript:**

L. 35: Switch order of "Interoperable", "Accessible"

*A: Done*

L. 151: This is not "hard to describe". Bringas and Goni did it, didn't they? Were their findings used, e.g. by correcting the fall rate equation to account for the initial velocity estimated from drop height? If so, where is this documented in the metadata? Or were their reported depth errors used in some sort of error estimate?

*A: Bringas and Goni (2015, BG15) carried out an accurate laboratory study in cylindrical tanks filled with water to describe the XBT motion after its impact with the sea surface as a function of the launch height. The BG15 algorithm corrects the depth value calculated by standard FRE when the launch height is different from about 3.0 - 4.0 m, but it is only an approximation of what happens in operational conditions that does not take into account the turbulence produced by the ship's wake, a perturbation that depends on the ship speed, its draft and the distance from the side of the hull. We found in the literature an unpublished communication from Gilson, Roemmich and Johnson (2008) that illustrates what happens when XBTs are launched from ships moving at different speeds, but it does not include any specific description about their behavior in the surface layer.*

*In our opinion, BG15 is a proper correction when the ship speed is close to zero, which is not the case for the majority of profiles in the REP dataset (Figure R1), so we preferred not to apply it to the REP dataset depth values and further investigate this issue in our next studies.*

*Figure R1 shows the distribution of launch heights per probe type in the REP dataset, indicating that about 70% of the drops were from platforms at ~ 10-11 m height.*

[Figure]

**Figure R1 - Distribution of XBT launch heights above the sea level per probe type in the REP dataset.**

L. 788: Change "point" to "profile"

*A: Done*

**Dataset:**

Something is wrong with the TEMPET01_TEST_QC variable. I assume it should encode, bitwise, the various QC tests. Assuming that there are <=16 tests, the variable type should then be a 16-bit integer (not a 64-bit float). The meaning of each bit should be explained in an attribute "flag_masks", not "flag_meanings", see http://cfconventions.org/cf-conventions/v1.6.0/cf-conventions.html#flags (section 3.5) for the difference between the two, and the values need to be re-computed (maybe I misunderstood, but the present values make no sense to me). In addition, it is unclear how these correspond to the values listed in table 2 of the manuscript, and in my data version, the content of flag_masks and flag_meanings have different lengths (15 vs. 13 entries).

*A: The variable TEMPET01_TEST_QC has been written as a 16-bit integer. Adopting the SDN convention, the QC variable has the mandatory attributes:*

- *flag_values: a list of all the flag values used in the encoding scheme*
- *flag_meanings: a list of the meanings associated with the codes in flag_values as space-delimited strings with internal spaces replaced by underscores.*

*Their different lengths depend on the fact that in the netcdf file the exit values 571, 572, 581, 582 are mentioned explicitly, while in Table 2 appear as 57# and 58#. We specified "(# = 1 or 2)" in Table 2.*

The present files have the 'cf_role' attribute assigned to the time variable. Since you actually have a variable "profile_id", I think this variable should have the cf_role attribute instead of time, or am I missing the logic here?

*A: The reviewer's suggestion is right, the new dataset has the cf_role assigned to profile_id.*

There are metadata entries that are presently global attributes, but they should be variables (or attributes) specific to each profile. These are factually incorrect at present: bathymetric_information, IMO_number, last_good_depth_according_to_operator, last_latitude_observation, last_longitude_observation, launching_height, max_acquisition_depth, max_recorded_depth, probe_manufacturer, probe_serial_number, recorder_types, ship_speed, fall_rate_equation_Coeff_1, fall_rate_equation_Coeff_2

I am unsure if this also applies to the following global attributes (please check): ices_platform_code, id, source_platform_category_code, sourceUrl, wmo_inst_type

*A: An url_metadata is associated to the entire dataset: each profile is identified through the corresponding profile_id and cruise_id variable. We inserted in Appendix A an example on how to retrieve these info.*

I recommend changing the dataset title (on the website and inside the files) exactly as follows (remove "of", correct "Tyrrhenian", and capitalize "Seas"): Reprocessed XBT dataset in the Ligurian and Tyrrhenian Seas (1999-2019)

*A: Done*

In the global attribute 'summary', correct spelling of "Expendible" to "Expendable"

*A: Done*

Reg. the fall rate coefficients:

They are presently given in global attributes, as if one set of coefficients applied to all probes. These change between probes; they have to be profile-specific.
The units are spelled wrong in the global attributes:
There needs to be a space between "m" and "s" (else, it is milliseconds)
The exponents behind "s" need to be negative

*A: Done*

The "coordinates" attributes are used incorrectly: DEPTH_INT is a coordinate and should not have such an attribute (I think, but correct me if I am wrong)

*A: We agree with the reviewer, only geophysical variables must have "coordinates" attribute as mandatory.*

TEMPET01_INT (and the other _INT variables except DEPTH_INT) should not list "depth" in the coordinates attribute, but rather "DEPTH_INT". This is important, because it defines the vertical position of the data - the way it is right now, you are telling the user that TEMPET01_INT data are coming from the wrong depth!

*A: We agree with the reviewer and we applied this suggestion.*

---

## Referee Report (RR1)

**General comments:**

The authors have much improved the manuscript since the first review. However, there are still some issues that both myself and the other reviewer highlighted that have not been fixed. These issues are associated with the data and metadata via ERDDAP and should be fixed before publication.

Despite the author's wish to follow FAIR principles, the data is not accessible in its current format as the entire dataset cannot be downloaded.

The attempt to make the metadata available using a separate set of url links in the main data file is not satisfactory. The effort that the authors have put into recovery of this information should lead to the metadata being included in the file as variables, not as a secondary data file only accessible via url coding.

One final suggestion is that the authors make the automatic QC code used available publicly, perhaps via a git repository.

**Manuscript specific comments:**

1. Line 210-217: The term 'accuracy' is used for the XCTDs. As these values are translated to 'uncertainty' in the data files, you need to explain that you have used the 'accuracy' values as type B uncertainty, as done in the Cowley et al uncertainty paper.

2. Line 375 – 379: Position on land check. I asked about this in the previous review and your response was: *There are not profiles are on land in the REP dataset, since the operators checked both the position and the launch time before the data transmission to the ENEA-STE. Since we did not encounter specific issues with date/time we did not implement additional checks.*
   I suggest you include this information in the paper to give the reader reassurance that there are no data flagged as bad due to position on land and that position errors were already fixed prior to this test.

3. Figure 10: please add more description to the figure caption so the reader doesn't have to refer back to the text. Or highlight individual issues with circles/shapes on the figure.

**Data comments:**

1. I downloaded a sub-setted netcdf file. It still has global attributes that should be removed from the global attributes section, and many need review. In particular, the green highlighted attributes belong to individual profiles and should be translated to variables. The yellow highlighted attributes are repeats of other attributes and include spelling mistakes.

```
// global attributes:
                :area = "Mediterranean Sea" ;
                :bathymetric_information = "102.0m" ;
                :cdm_data_type = "Profile" ;
                :cdm_profile_variables =
"latitude,longitude,time,profile_id" ;
                :citation = "Reseghetti F., Fratianni C., Simoncelli S.
(2024). Reprocessed of XBT dataset in the Ligurian and Tyrrhenian seas
```

(1999-2019) (Version 2) [Data set]. Istituto Nazionale di Geofisica e
Vulcanologia (INGV). https://doi.org/10.13127/rep_xbt_1999_2019.2" ;
                :contributor_name = "Franco Reseghetti, Claudia Fratianni,
Simona Simoncelli, Antonio Guarnieri" ;
                :contributor_role = "Data Collector, Data Curator/Manager,
Researcher/Supervisor, Project Leader" ;
                :contributor_url = "orcid.org/0000-0002-7569-8541,
orcid.org/0000-0002-4983-4255, orcid.org/0000-0003-1283-2798,
orcid.org/0000-0001-8162-6571" ;
                :Conventions = "SeaDataNet_1.0, CF-1.6, COARDS, ACDD-1.3" ;
                :creator_name = "Istituto Nazionale di Geofisica e
Vulcanologia, Sede di Bologna (INGV)" ;
                :creator_url = "https://www.ingv.it/" ;
                :data_mode = "D" ;
                :depth_uncertainty = "depth <=230m: 4.6m; depth >=230m: 2%
(Table 2 from Cowley R et al., 2021
https://doi.org/10.3389/fmars.2021.689695)" ;
                :doi = "https://doi.org/10.13127/rep_xbt_1999_2019.2" ;
                :Easternmost_Easting = 9.1445 ; Repeat of the geospatial
attributes
                :fall_rate_equation_Coeff_1 = "6.301m s^-1" ;
                :fall_rate_equation_Coeff_2 = "-0.00216m s^-2" ;
                :family_code = "XB" ;
                :family_label = "vessel" ;
                :featureType = "Profile" ;
                :geospatial_lat_max = 44.2966 ;
                :geospatial_lat_min = 43.8832 ;
                :geospatial_lat_units = "degrees_north" ;
                :geospatial_lon_max = 9.1445 ;
                :geospatial_lon_min = 8.8679 ;
                :geospatial_lon_units = "degrees_east" ;
                :geospatial_vertical_max = 550.59f ;
                :geospatial_vertical_min = 0.f ;
                :geospatial_vertical_positive = "down" ;
                :geospatial_vertical_units = "m" ;
                :history = "2024-07-08T02:56:09Z (local files)\n",
                      "2024-07-08T02:56:09Z
http://oceano.bo.ingv.it/erddap/tabledap/REP_XBT_1999_2019_v2.nc?CALIB%2Cde
pth%2CDEPTH_COR%2CDEPTH_COR_FLAGS_QC%2CDEPTH_COR_INT%2CDEPTH_COR_INT_SEADAT
ANET_QC%2CDEPTH_FLAGS_QC%2CDEPTH_INT%2CDEPTH_INT_SEADATANET_QC%2CDEPTH_TEST
_QC%2Clatitude%2Clongitude%2CPOSITION_SEADATANET_QC%2CSDN_BOT_DEPTH%2CSDN_C
RUISE%2CSDN_EDMO_CODE%2CTEMPET01%2CTEMPET01_COR%2CTEMPET01_COR_FLAGS_QC%2CT
EMPET01_COR_INT%2CTEMPET01_COR_INT_SEADATANET_QC%2CTEMPET01_FLAGS_QC%2CTEMP
ET01_INT%2CTEMPET01_INT_SEADATANET_QC%2CTEMPET01_TEST_QC%2Ctime%2CTIME_SEAD
ATANET_QC%2Ccruise_id%2Cprofile_id%2Curl_metadata&time%3E=2019-09-
21T00%3A00%3A00Z&time%3C=2019-09-28T00%3A21%3A13Z" ;
                :ices_platform_code = "48AA" ;
                :id = "T0_M1947" ;
                :IMO_number = "8642751" ;
                :infoUrl = "https://www.enea.it/it//,http://www.ingv.it" ;
                :institution = "ENEA Centro Ricerche Ambiente Marino - La
Spezia, Istituto Nazionale di Geofisica e Vulcanologia, Sede di Bologna
(INGV)" ;
                :institution_edmo_code = "136, 251" ;
                :keywords = "1999-2019, ambiente, applied, applying,
barker, bathymetric, bathymetry, bathythermograph, below, bologna, CALIB,
calibrated, calibration, cdi, centro, cheng2014, code, control, cor,
corrected, cruise, cruise_id, data, dataset, depth, DEPTH_COR,
DEPTH_COR_FLAGS_QC, DEPTH_COR_INT, DEPTH_COR_INT_SEADATANET_QC,
DEPTH_FLAGS_QC, DEPTH_INT, DEPTH_INT_SEADATANET_QC, DEPTH_TEST_QC,
directory, each, earth, Earth Science > Oceans > Bathymetry/Seafloor

Topography > Bathymetry, Earth Science > Oceans > Ocean Temperature > Water
Temperature, earth science>oceans>ocean temperature>ocean temperature
profiles, enea, european, expendible, flag, flags, floor, geofisica,
grouping, ingv, interpolated, istituto, label, latitude, levels, ligurian,
longitude, MACMAP, mapping, marine, marino, mcdougall, measurement, method,
nazionale, ocean, oceans, organizations, original, partner,
POSITION_SEADATANET_QC, profile, profile_id, quality, rep_xbt_1999_2019_v2,
reprocessed, resolution, resulting, ricerche, science, SDN_BOT_DEPTH,
SDN_CRUISE, SDN_EDMO_CODE, sea, sea_floor_depth_below_sea_surface,
sea_water_temperature, seadatanet, seafloor, seas, seawater, sede, site,
spezia, standard, status, status_flag, surface, temperature, tempet01,
TEMPET01_COR, TEMPET01_COR_FLAGS_QC, TEMPET01_COR_INT,
TEMPET01_COR_INT_SEADATANET_QC, TEMPET01_FLAGS_QC, TEMPET01_INT,
TEMPET01_INT_SEADATANET_QC, TEMPET01_TEST_QC, test, thyrrhenian, time,
TIME_SEADATANET_QC, topography, using, vertical, vulcanologia, water, xbt"
;
    :keywords_vocabulary = "GCMD Science Keywords" ;
    :last_good_depth_according_to_operator = "92.78m" ;
    :last_latitude_observation = "44.2966" ;
    :last_longitude_observation = "9.1445" ;
    :launching_height = "05m" ;
    :license = "Attribution (CC BY 4.0)" ;
    :max_acquisition_depth = "300m" ;
    :max_recorded_depth = "292m" ;
    :netcdf_version = "netCDF-4 classic model" ;
    :Northernmost_Northing = 44.2966 ; Repeat of the geospatial
attributes
    :pi_name = "Franco Reseghetti" ;
    :platform_code = "IGMA" ;
    :platform_name = "Ammiraglio Magnaghi" ;
    :probe_manufacturer = "Lockheed Martin Sippican" ;
    :probe_serial_number = "345751" ;
    :probe_type = "T-10" ;
    :qc_indicator = "excellent (all important QC done)" ;
    :recorder_sampling_frequency = "10Hz" ;
    :recorder_serial_number = "239" ;
    :recorder_types = "72" ;
    :ship_speed = "06kn" ;
    :source_platform_category_code = "31" ;
    :sourceUrl = "(local files)" ;
    :Southernmost_Northing = 43.8832 ; Repeat of the geospatial
attributes
    :standard_name_vocabulary = "CF Standard Name Table v70" ;
    :subsetVariables = "CALIB, latitude, longitude,
POSITION_SEADATANET_QC, SDN_BOT_DEPTH, SDN_CRUISE, SDN_EDMO_CODE, time,
TIME_SEADATANET_QC, cruise_id, profile_id" ;
    :summary = "The Rep_XBT_1999_2019 (V2) dataset comes from
an improved and fully automatic quality control procedure applied to all
available raw data and metadata from Expendable Bathytermograph (Expendible
Bathythermograph (XBT)) probes sampled and managed by ENEA S.Teresa Centre
since September 1999 in the Ligurian and Thyrrhenian Seas. The reprocessed
dataset contains a full metadata description obtained from the cruise
reports according to the most recent community standards and formats. The
second version of the dataset has been released as a final result of the
revised manuscript" ;
    :TEMPET01_uncertainty = "XBT = 0.10 deg C; XCTD = 0.02 deg
C  (Table 2 from Cowley R et al., 2021
https://doi.org/10.3389/fmars.2021.689695)" ;
    :time_coverage_end = "2019-09-28T00:21:13Z" ;
    :time_coverage_start = "2019-09-27T20:00:22Z" ;

```
                    :title = "Reprocessed XBT dataset (V2) in the Ligurian and
Thyrrhenian Seas (1999-2019)" ;
                    :update_interval = "void" ; Is this redundant, if so
remove?
                    :Westernmost_Easting = 8.8679 ; Repeat of the geospatial
attributes
                    :wmo_inst_type = "061" ;
                    :wmo_platform_code = "IGMA" ;
```

2. Thanks for including uncertainties. I suggest 'depth_uncertainty' should be moved to an attribute in the Depth* variables or calculated out and added as its own variable. 'TEMPET01_uncertainty' should also be an attribute in the TEMPET01 variable and be specific to the instrument used (XBT or CTD). The reference can still be included as a global attribute.

3. The url_metadata link should be replaced with the actual data. To use the links, more coding is required (thanks for the example in C.6). Each link goes to a single line of information on a html page that could just as easily be included as variables in the datafile. I strongly suggest you put the information in these links into variables, especially since one of the main purposes of the paper is to rescue metadata and ensure it is attached to the data (see your own introduction), as well as make it Accessible.

4. Attempts to download the full dataset as netcdf, csv or txt format failed due to the size of the dataset – is this a server-side issue? The error is:

```
Error {
    code=413;
    message="Payload Too Large: Your query produced too much data.  Try to
request less data. [memory]  The request needs more memory (80337 MB) than
is ever safely available in this Java setup (24072 MB).
(TableWriterAll.cumulativeTable)";
}
```

5. In an attempt to get access to the full dataset, I tried using the url generated in ERDDAP in Matlab and Python.
```
http://oceano.bo.ingv.it/erddap/tabledap/REP_XBT_1999_2019_v2.nc?CALIB%2Cde
pth%2CDEPTH_COR%2CDEPTH_COR_FLAGS_QC%2CDEPTH_COR_INT%2CDEPTH_COR_INT_SEADAT
ANET_QC%2CDEPTH_FLAGS_QC%2CDEPTH_INT%2CDEPTH_INT_SEADATANET_QC%2CDEPTH_TEST
_QC%2Clatitude%2Clongitude%2CPOSITION_SEADATANET_QC%2CSDN_BOT_DEPTH%2CSDN_C
RUISE%2CSDN_EDMO_CODE%2CTEMPET01%2CTEMPET01_COR%2CTEMPET01_COR_FLAGS_QC%2CT
EMPET01_COR_INT%2CTEMPET01_COR_INT_SEADATANET_QC%2CTEMPET01_FLAGS_QC%2CTEMP
ET01_INT%2CTEMPET01_INT_SEADATANET_QC%2CTEMPET01_TEST_QC%2Ctime%2CTIME_SEAD
ATANET_QC%2Ccruise_id%2Cprofile_id%2Curl_metadata&distinct()
```

If used as is, it crashed both Matlab and Python. After getting to the end of the paper, there is an example of python code to access the metadata for a single profile. A similar example of how to access the entire dataset with the url would be useful as the ERDDAP url above might need to be adjusted to work.

6. Spelling errors in TEMPET01_TEST_QC variable attribute flag_meanings:

`negative_vertical_gradient_at_`==`firts`==`_iteration and`
`positive_vertical_gradient_at_`==`firts`==`_iteration`

Also, the following flag_meanings do not match with the ones described in the paper and the flag_values are not included in the list (there are more flag_ meanings than flag_values, as mentioned by Reviewer 2 in the first review).

`negative_vertical_gradient_at_third_iteration`
`positive_vertical_gradient_at_third_iteration`

7. In the first review I asked: Is the TEMPET01_TEST_QC variable additive?
Your reply didn't address the issue (and I think Reviewer 2 asked the same question and described it better). Let me try again.
There are several tests applied independently to each profile. If a single temperature fails more than one test, which exit value do you use? Or, do you record all the exit values for that data point and if you do, are they added together bit-wise as described in the netcdf conventions here: https://cfconventions.org/cf-conventions/cf-conventions.html#flag-variable-flag-masks-flag-values-ex
The way you are presenting the exit values does not meet the CF conventions and if more than one test is failed for a data point (as shown in your Figure 2 where both gross check and wire-stretch are failed for the same data points), how is that represented?

---

## Author Response (AR2)

**General comments:**

The authors have much improved the manuscript since the first review. However, there are still some issues that both myself and the other reviewer highlighted that have not been fixed. These issues are associated with the data and metadata via ERDDAP and should be fixed before publication.

Despite the author's wish to follow FAIR principles, the data is not accessible in its current format as the entire dataset cannot be downloaded.

The attempt to make the metadata available using a separate set of url links in the main data file is not satisfactory. The effort that the authors have put into recovery of this information should lead to the metadata being included in the file as variables, not as a secondary data file only accessible via url coding.

One final suggestion is that the authors make the automatic QC code used available publicly, perhaps via a git repository.

Answer:
We thank the reviewer for the detailed review which allows us to further improve the dataset and make it fit for use. This is crucial for us that decided to adopt ERDDAP server, a FAIR-compliant data access service (*O'Brien and Delaney, 2024*), in line with the GOOS (Global Ocean Observing System) Observations Coordination Group (https://goosocean.org/who-we-are/observations-coordination-group/) strategy. In fact, according to *Lange et al. (2023)*, ERDDAP "*(i) supports dozens of popular formats; (ii) provides standards-based metadata and data services and formats; (iii) supports federated access of distributed ERDDAP data services; (iv) supports both human and machine interactions; (v) supports sub-setting of large datasets; (vi) provides improved discovery of datasets through commercial search engines; and (vii) provides support for archival of datasets*". (Text added to Section 7)

We are sorry that the data was not accessible, we have tried several times since your feedback and we had no problems in downloading the data. We hypothesize an incidental server load and traffic.

We are also sorry that the availability of all existing metadata as a URL link is not satisfactory for the reviewer. The decision to provide metadata in a separate metadata URL has been taken in collaboration with the EMODnet Physics data managers within the framework of EMODnet Data Ingestion project (*Novellino et al., 2024*). In fact, the REP XBT dataset is also accessible through EMODnet Physics data portal, thanks to the ERDDAP machine to machine interoperability. Including all metadata as variables in the file, would make the ERDDAP conversion to the dozens output formats less efficient due to the many metadata strings to manage.

To meet the reviewer request and facilitate the data reusability we prepared a Jupyter Notebook in Python that allows recombining all data and metadata in NetCDF files, one per XBT profile. The notebook is available on a GitHub repository and published on Zenodo, in compliance with Open Science principles.

The publication of the automatic QC code was out of our scope when reprocessing and analyzing the XBT data, so we did not work to prepare a FAIR software (*Baker et al., 2022*) but to prepare a FAIR dataset. We believe that sharing a code which is not fit-for-use at this stage of the review

process would not be beneficial for both producers and users. We would like, if accepted the dataset paper, to work on optimizing the code and making it FAIR. We think that publishing it in an open software journal could help to improve it by going through a peer review.

*Barker, M., Chue Hong, N.P., Katz, D.S. et al. Introducing the FAIR Principles for research software. Sci Data 9, 622 (2022). https://doi.org/10.1038/s41597-022-01710-x.*

*Lange N, Tanhua T, Pfeil B, Bange HW, Lauvset SK, Grégoire M, Bakker DCE, Jones SD, Fiedler B, O'Brien KM and Körtzinger A (2023) A status assessment of selected data synthesis products for ocean biogeochemistry. Front. Mar. Sci. 10:1078908. doi: 10.3389/fmars.2023.1078908*

*Novellino, A., Pizziol, V., Dapueto, G., Misurale, F., Scotto, B. M., Bordoni, R., Gorringe, P., Schaap, D., & Iona, A. (2024). EMODnet Ingestion and the operational data exchange examples and hot topics. Miscellanea INGV, 80, 364–366. https://doi.org/10.13127/MISC/80/140*

*O'Brien, K., & Delaney, C. (2024). A review of ERDDAP the established best practice in sharing gridded and tabular data from the Earth Sciences community. Miscellanea INGV, 80, 231–232. https://doi.org/10.13127/MISC/80/87*

**Manuscript specific comments:**

Line 210-217: The term 'accuracy' is used for the XCTDs. As these values are translated to 'uncertainty' in the data files, you need to explain that you have used the 'accuracy' values as type B uncertainty, as done in the Cowley et al uncertainty paper.

Answer: Text has been added: "*These accuracy values can be considered Type B uncertainties, as in Cowley et al. (2021), and they are included in the REP dataset metadata information.*"

Line 375–379: Position on land check. I asked about this in the previous review and your response was: "*There are no profiles on land in the REP dataset, since the operators checked both the position and the launch time before the data transmission to the ENEA-STE. Since we did not encounter specific issues with date/time we did not implement additional checks.*" I suggest you include this information in the paper to give the reader reassurance that there are no data flagged as bad due to position on land and that position errors were already fixed prior to this test.

Answer: We thank the reviewer for this suggestion, we added a phrase in the manuscript.

Figure10: please add more description to the figure caption so the reader doesn't have to refer back to the text. Or highlight individual issues with circles/shapes on the figure.

Answer: We thank the reviewer for the suggestion, some text has been added to the caption and some shapes have been added on the plots. Some text has been eliminated at page 26 too.

**Data comments:**

- I downloaded a sub-setted netcdf file. It still has global attributes that should be removed from the global attributes section, and many need review. In particular, the green highlighted attributes belong to individual profiles and should be translated to variables. The yellow highlighted attributes are repeats of other attributes and include spelling mistakes.

Answer: The metadata reported by the reviewer have been subdivided into blocks to manage specific answers.

Spelling mistakes have been corrected.

```
// global attributes:
    :area = "Mediterranean Sea" ;
    :bathymetric_information = "102.0m" ; :cdm_data_type = "Profile" ;
    :cdm_profile_variables = "latitude,longitude,time,profile_id" ;
    :citation = "Reseghetti F., Fratianni C., Simoncelli S. (2024). Reprocessed of XBT dataset in the Ligurian and Tyrrhenian seas (1999-2019) (Version 2) [Data set]. Istituto Nazionale di Geofisica e Vulcanologia (INGV). https://doi.org/10.13127/rep_xbt_1999_2019.2";
    :contributor_name = "Franco Reseghetti, Claudia Fratianni, Simona Simoncelli, Antonio Guarnieri" ;
    :contributor_role = "Data Collector, Data Curator/Manager, Researcher/Supervisor, Project Leader" ;
    :contributor_url = "orcid.org/0000-0002-7569-8541, orcid.org/0000-0002-4983-4255, orcid.org/0000-0003-1283-2798, orcid.org/0000-0001-8162-6571";
    :Conventions = "SeaDataNet_1.0, CF-1.6, COARDS, ACDD-1.3";
    :creator_name = "Istituto Nazionale di Geofisica e Vulcanologia, Sede di Bologna (INGV)" ;
    :creator_url = "https://www.ingv.it/" ;
    :data_mode = "D" ;
    :depth_uncertainty = "depth <=230m: 4.6m; depth >=230m: 2% (Table 2 from Cowley R et al., 2021 https://doi.org/10.3389/fmars.2021.689695)";
    :doi = "https://doi.org/10.13127/rep_xbt_1999_2019.2";
    :feature_Type = "Profile";
    :history = "2024-07-08T02:56:09Z (local files)\n" "2024-07-08T02:56:09Z http://oceano.bo.ingv.it/erddap/tabledap/REP_XBT_1999_2019_v2.nc%CALIB%2Cdepth%2CDEPTH_COR%2CDEPTH_COR_FLAGS_QC%2CDEPTH_COR_INT%2CDEPTH_COR_INT_SEADAT
    ANET_QC%2CDEPTH_FLAGS_QC%2CDEPTH_INT%2CDEPTH_INT_SEADATANET_QC%2CDEPTH_TEST_QC%2Clatitude%2Clongitude%2CPOSITION_SEADATANET_QC%2CSDN_BOT_DEPTH%2CSDN_C RUISE%2CSDN_EDMO_CODE%2CTEMPET01%2CTEMPET01_COR%2CTEMPET01_COR_FLAGS_QC%2CT
    EMPET01_COR_INT%2CTEMPET01_COR_INT_SEADATANET_QC%2CTEMPET01_FLAGS_QC%2CTEMP ET01_INT%2CTEMPET01_INT_SEADATANET_QC%2CTEMPET01_TEST_QC%2Ctime%2CTIME_SEAD ATANET_QC%2Ccruise_id%2Cprofile_id%2Curl_metadata&time%3E=2019-09-21T00%3A00%3A00Z&time%3C=2019-09-
    28T00%3A21%3A13Z" ;
    :infoUrl = "https://www.enea.it/it/,http://www.ingv.it/" ;
    :institution = "ENEA Centro Ricerche Ambiente Marino - La Spezia, Istituto Nazionale di Geofisica e Vulcanologia, Sede di Bologna (INGV)" ;
    :keywords = "1999-2019, ambiente, applied, applying, barker, bathymetric, bathymetry, bathythermograph, below, bologna, CALIB, calibrated, calibration, cdi, centro, cheng2014, code, control, cor, corrected, cruise, cruise_id, data, dataset, depth, DEPTH_COR, DEPTH_COR_FLAGS_QC, DEPTH_COR_INT, DEPTH_COR_INT_SEADATANET_QC,
    DEPTH_FLAGS_QC, DEPTH_INT, DEPTH_INT_SEADATANET_QC, DEPTH_TEST_QC, directory, each, earth, Earth Science > Oceans > Bathymetry/Seafloor Topography > Bathymetry, Earth Science > Oceans > Ocean Temperature > Water Temperature, earth science>oceans>ocean temperature>ocean temperature profiles, enea, european, expendible, flag,
    flags, floor, geofisica, grouping, ingv, interpolated, istituto, label, latitude, levels, ligurian, longitude, MACMAP, mapping, marine, marino, mcdougall, measurement, method, nazionale, ocean, oceans, organizations, original, partner, POSITION_SEADATANET_QC, profile, profile_id, quality, rep_xbt_1999_2019_v2, reprocessed, resolution, resulting,
    ricerche, science, SDN_BOT_DEPTH, SDN_CRUISE, SDN_EDMO_CODE, sea, sea_floor_depth_below_sea_surface, sea_water_temperature, seadatanet, seafloor, seas, seawater, sede, site, spezia, standard, status, status_flag, surface, temperature, tempet01, TEMPET01_COR, TEMPET01_COR_FLAGS_QC, TEMPET01_COR_INT,
    TEMPET01_COR_INT_SEADATANET_QC, TEMPET01_FLAGS_QC, TEMPET01_INT, TEMPET01_INT_SEADATANET_QC, TEMPET01_TEST_QC, test, thyrrhenian, time, TIME_SEADATANET_QC, topography, using, vertical, vulcanologia, water, xbt" ;
    :keywords_vocabulary = "GCMD Science Keywords" ;
    netcdf_version = "netCDF-4 classic model" ;
    :pi_name = "Franco Reseghetti" ;
    :sourceUrl = "(local files)" ;
    :standard_name_vocabulary = "CF Standard Name Table v70" ;
    :subsetVariables = "CALIB, latitude, longitude, POSITION_SEADATANET_QC, SDN_BOT_DEPTH, SDN_CRUISE, SDN_EDMO_CODE, time, TIME_SEADATANET_QC, cruise_id, profile_id" ;
    :summary = "The Rep_XBT_1999_2019 (V2) dataset comes from an improved and fully automatic quality control procedure applied to all available raw data and metadata from Expendable Bathytermograph (XBT) probes sampled and managed by ENEA S.Teresa Centre since September 1999 in the Ligurian and Thyrrhenian Seas. The reprocessed dataset
    contains a full metadata description obtained from the cruise reports according to the most recent community standards and formats. The second version of the dataset has been released as a final result of the revised manuscript" ;
    :TEMPET01_uncertainty = "XBT = 0.10 deg C; XCTD = 0.02 deg C (Table 2 from Cowley R et al., 2021 https://doi.org/10.3389/fmars.2021.689695)" ;
    :title = "Reprocessed XBT dataset (V2) in the Ligurian and Thyrrhenian Seas (1999-2019)" ;
    :license = "Attribution (CC BY 4.0)" ;
```

The below attributes, referring to each single profile, have been removed from global attributes and included in the url_metadata, which is a variable URL linking to the metadata information. Data and metadata of each profile can be easily associated through the *profile_id* and *cruise_id* fields.

```
:fall_rate_equation_Coeff_1 = "6.301m s^-1" ;
:fall_rate_equation_Coeff_2 = "0.00216m s^-2";
:family_code = "XB";
:ices_platform_code = "48AA";
:IMO_number = "8642751";
:launching_height = "05m";
:last_good_depth_according_to_operator = "92.78m";
:max_acquisition_depth = "300m";
:max_recorded_depth = "292m";
:platform_code = "IGMA";
:platform_name = "Ammiraglio Magnaghi";
:probe_manufacturer = "Lockheed Martin Sippican";
:probe_serial_number = "345751";
:probe_type = "T-10";
:qc_indicator = "excellent (all important QC done)";
:recorder_sampling_frequency = "10Hz";
:recorder_serial_number = "239";
:recorder_types = "72";
:ship_speed = "06kn";
:source_platform_category_code = "31";
:wmo_inst_type = "061" ;
:wmo_platform_code = "IGMA" ;
```

We decided to remove this information from global attributes since it is referring to internal identification of the profile:
```
:id = "T0_M1947";
```

The metadata attributes below have been removed since they are repeats of geospatial attributes.
```
:last_latitude_observation = "44.2966"
:last_longitude_observation = "9.1445"
```

The global metadata attributes below, refer to the entire dataset or to a portion of the selected dataset. The green ones are kept since they are part of the recommended Attribute Convention for Data Discovery 1-3 (ACDD). The geospatial yellow attributes are inserted by default in ERDDAP global metadata attributes to visualize the data via Google Earth, thus they cannot be removed.
```
:geospatial_lat_max = 44.2966 ;
:geospatial_lat_min = 43.8832 ;
:geospatial_lon_max = 9.1445 ;
:geospatial_lon_min = 8.8679 ;
```

```
:geospatial_vertical_max = 550.59f ;
:geospatial_vertical_min = 0.f ;
:time_coverage_end = ""2019-09-28T00:21:13Z"" ;
:time_coverage_start = ""2019-09-27T20:00:22Z"" ;
:Easternmost_Easting = 9.1445 ;
:Northernmost_Northing = 44.2966 ;
:Southernmost_Northing = 43.8832 ;
:Westernmost_Easting = 8.8679 ;
```

This is a ACDD global attribute referring to Publication information. It is not required but we decided to include it since it gives information about the dataset update. In our case it is "void" because the dataset is not updated on a schedule.
```
:update_interval = ""void"" ;
```

- Thanks for including uncertainties. I suggest 'depth_uncertainty' should be moved to an attribute in the Depth* variables or calculated out and added as its own variable. 'TEMPET01_uncertainty' should also be an attribute in the TEMPET01 variable and be specific to the instrument used (XBT or CTD). The reference can still be included as a global attribute.

Answer: As suggested by the reviewer, the uncertainty info has been moved to depth and TEMPET01 variables.

- The url_metadata link should be replaced with the actual data. To use the links, more coding is required (thanks for the example in C.6). Each link goes to a single line of information on a html page that could just as easily be included as variables in the datafile. I strongly suggest you put the information in these links into variables, especially since one of the main purposes of the paper is to rescue metadata and ensure it is attached to the data (see your own introduction), as well as make it Accessible.

Answer: The url_metadata is a URL variable that we provide in agreement with the EMODnet Physics data managers. In fact, EMODnet Physics portal distributes our dataset through a federated ERDDAP server's approach permitted by machine to machine interoperability. The practice of defining an *url_metadata* is advisable to manage more efficiently the numerous strings that are used for metadata information.
We decided to provide a Jupyter Notebook in Python (*Fratianni and Frizzera, 2024*) that allows combining data and metadata into a single file to facilitate data reusability and be compliant with the FAIR data principles, as stated in the manuscript.
The notebook is available on a GitHub repository and published on Zenodo at https://doi.org/10.5281/zenodo.13862792. We have modified the annex C.6 to include this change.

*Fratianni, C., & Frizzera, P. (2024). REPROCESSED XBT 1999-2019: how to access data and metadata throught ERDDAP (v1.0.0). Zenodo. https://doi.org/10.5281/zenodo.13862792*

- Attempts to download the full dataset as netcdf, csv or txt format failed due to the size of the dataset – is this a server-side issue? The error is:
```
Error {
    code=413;
    message="Payload Too Large: Your query produced too much data. Try to request less data.
[memory] The request needs more memory (80337 MB) than is ever safely available in this Java
setup (24072 MB). (TableWriterAll.cumulativeTable)";
}
```

Answer: We tried several times to download the entire dataset, both by selecting via browser interface the entire period and download as netcdf, csv and text format and via provided python script: we didn't have any issue. We are supposing that the encountered issue could be ascribed to incidental server load and traffic.

- In an attempt to get access to the full dataset, I tried using the url generated in ERDDAP in Matlab and Python.

```
http://oceano.bo.ingv.it/erddap/tabledap/REP_XBT_1999_2019_v2.nc?CALIB%2Cde
pth%2CDEPTH_COR%2CDEPTH_COR_FLAGS_QC%2CDEPTH_COR_INT%2CDEPTH_COR_INT_SEADAT
ANET_QC%2CDEPTH_FLAGS_QC%2CDEPTH_INT%2CDEPTH_INT_SEADATANET_QC%2CDEPTH_TEST
_QC%2Clatitude%2Clongitude%2CPOSITION_SEADATANET_QC%2CSDN_BOT_DEPTH%2CSDN_C
RUISE%2CSDN_EDMO_CODE%2CTEMPET01%2CTEMPET01_COR%2CTEMPET01_COR_FLAGS_QC%2CT
EMPET01_COR_INT%2CTEMPET01_COR_INT_SEADATANET_QC%2CTEMPET01_FLAGS_QC%2CTEMP
ET01_INT%2CTEMPET01_INT_SEADATANET_QC%2CTEMPET01_TEST_QC%2Ctime%2CTIME_SEAD
ATANET_QC%2Ccruise_id%2Cprofile_id%2Curl_metadata&distinct()
```

If used as is, it crashed both Matlab and Python. After getting to the end of the paper, there is an example of python code to access the metadata for a single profile. A similar example of how to access the entire dataset with the url would be useful as the ERDDAP url above might need to be adjusted to work.

Answer: We have prepared a Jupyter Notebook (*Fratianni, C. and Frizzera, P., 2024*), as suggested by the reviewer, to facilitate the access to the entire dataset and the recombination of data and metadata. The notebook is available on a GitHub repository and published on Zenodo at https://doi.org/10.5281/zenodo.13862792. We have modified Section 7 and annex C.6 to include this change.

*Fratianni, C., & Frizzera, P. (2024). REPROCESSED XBT 1999-2019: how to access data and metadata throught ERDDAP (v1.0.0). Zenodo. https://doi.org/10.5281/zenodo.13862792*

- Spelling errors in TEMPET01_TEST_QC variable attribute flag_meanings:
```
negative_vertical_gradient_at_ _iteration and positive_vertical_gradient_at_
_iteration
```
  Also, the following flag_meanings do not match with the ones described in the paper and the flag_values are not included in the list (there are more flag_ meanings than flag_values, as mentioned by Reviewer 2 in the first review).
```
negative_vertical_gradient_at_third_iteration
positive_vertical_gradient_at_third_iteration
```

Answer: We corrected spelling mistakes and we also provided the corresponding flag meanings and flag values.

- In the first review I asked: Is the TEMPET01_TEST_QC variable additive? Your reply didn't address the issue (and I think Reviewer 2 asked the same question and described it better). Let me try again. There are several tests applied independently to each profile. If a single temperature fails more than one test, which exit value do you use? Or, do you record all the exit values for that data point and if you do, are they added together bit-wise as described in the netcdf conventions here: https://cfconventions.org/cf- conventions/cf-conventions.html#flag-variable-flag-masks-flag-values-ex. The way you are presenting the exit values does not meet the CF conventions and if more than one test is failed for a data point (as shown in your Figure 2 where both gross check and wire-stretch are failed for the same data points), how is that represented?

Answer: We thank the reviewer for stressing this point, we realize that the manuscript is not clearly describing how the QC tests are documented in the metadata. All the QC tests performed and their results (Table 2) are documented in three different ancillary variables: POSITION_SEADATANET_QC, DEPTH_TEST_QC, TEMPET01_TEST_QC. We slightly modified the text at the beginning of Section 4.1 and we added a column in Table 2.

None of the ancillary variables (POSITION_SEADATANET_QC, DEPTH_TEST_QC, TEMPET01_TEST_QC) is additive, we record each test exit value. We adopted the SeaDataNet convention and not the CF convention.

➔ The POSITION_SEADATANET_QC is defined by a single test (test 1 in Table 2).
➔ The DEPTH_TEST_QC contains the outcome of two tests, one based on GEBCO local bathymetry (test 2 in Table 2) and one based on the last good depth recorded by the operator (test 3 in Table 2). Since the GEBCO local bathymetry was often in disagreement with the operator information we decided to keep the output of test 3 in DEPTH_FLAGS_QC, the ancillary variable containing the resulting quality flags associated to each record in the depth profile (text has been added to Sec. 4.2).
➔ The TEMPET01_TEST_QC contains the exit values of six tests (tests 4-9) and Section 4.2 explains how we mapped the test exit values to the TEMPET01_FLAGS_QC. An example of the TEMPET01_TEST_QC (6 columns per a number of rows corresponding to the profile records) and the resulting TEMPET01_FLAGS_QC is reported below:

```
TEMP01_TEST_QC =
    49   50   48   48   48   49
    49   50   48   48   48   49
    49   49   48   48   48   49
    49   49   48   48   48   49
    49   49   48   48   48   49
    49   49   48   48   48   49
    49   48   49   49   49   49
    49   48   49   49   49   49
    49   48   49   49   49   49
    49   48   49   49   49   49
    49   48   56   49   49   49
    49   48   49   49   49   49
    ...

TEMP01_FLAGS_QC =

    ...
```

We slightly modified Section 4.2 to clarify the issues raised by the reviewer.